# ACTOR-CRITIC ALIGNMENT FOR OFFLINE-TO-ONLINE REINFORCEMENT LEARNING

## ABSTRACT

Deep offline reinforcement learning has recently demonstrated considerable promise in leveraging offline datasets, providing high-quality models that significantly reduce the online interactions required for fine-tuning. However, such a benefit is often diminished due to the marked state-action distribution shift, which causes significant bootstrap error and wipes out the good initial policy. Existing solutions resort to constraining the policy shift or balancing the sample replay based on their online-ness. However, they require online estimation of distribution divergence or density ratio. To avoid such complications, we propose deviating from existing actor-critic approaches that directly transfer the state-action value functions. Instead, we post-process them by aligning with the offline learned policy, so that the $Q$-values for actions *outside* the offline policy are also tamed. As a result, the online fine-tuning can be simply performed as in the standard actor-critic algorithms. We show empirically that the proposed method improves the performance of the fine-tuned robotic agents on various simulated tasks.

## 1 INTRODUCTION

Offline reinforcement learning (RL) provides a novel tool that allows offline batch data to be leveraged by RL algorithms without having to interact with the environment (Levine et al., 2020). This opens up new opportunities for important scenarios such as health care decision making, and goal-directed dialog learning. Due to the limitation of offline data, it generally remains beneficial and necessary to fine-tune the learned model through online interactions, and ideally the latter will enjoy a faster learning curve thanks to the favorable initialization.

Unfortunately, it has been long observed that a direct offline-to-online (O2O) transfer often leads to catastrophic degradation of performance in the online stage, which is unacceptable in critical applications including medical treatment and autonomous driving. A key cause lies in the significant shift of state distribution at online phase compared with the offline data (Fujimoto et al., 2019; Kumar et al., 2019; Fu et al., 2019; Kumar et al., 2020a). As a result, the Bellman backup suffers a compounded error (Farahmand et al., 2010; Munos, 2005), because the $Q$-value has not been well estimated for the state-actions lying outside the offline distribution.

A number of solutions have been developed to address this issue. The most straightforward approach is importance sampling (Laroche et al., 2019; Gelada & Bellemare, 2019; Zhang et al., 2020; Huang & Jiang, 2020), which requires an additional effort of estimating the behavior policy, and suffers from high variance, especially when it differs markedly from the learned policy (a more realistic issue for the offline setting than the conventional off-policy setting). The model-based approach, on the other hand, also suffers from the distribution shift in state marginals and actions (Mao et al., 2022; Kidambi et al., 2020; Yu et al., 2020; Janner et al., 2019). It may exploit the model to pursue out-of-distribution states and actions where the model mis-believes to yield a high return. So they also require detecting and quantifying the shift. In addition, they suffer from standard challenges plaguing model-based RL algorithms such as long horizon and high dimensionality.

Dynamic programming proffers lower variance and directly learns the value functions and policy. Several approaches have been proposed to combat distribution shift. A natural idea is to constrain the policy to the proximity of the behavior policy, and this has been implemented by using probability divergences in (Nair et al., 2020; Siegel et al., 2020; Peng et al., 2019; Wu et al., 2019; Kumar et al., 2019), or by behavior cloning regularization (Zhao et al., 2021; Fujimoto & Gu, 2021). A

second class of approaches resort to pessimistic under-estimate of the state-action values (Kumar et al., 2020b; Kostrikov et al., 2021), especially for out-of-distribution actions that could have an unjustified high value. Conservative Q-learning (CQL, Kumar et al., 2020b) has been shown to produce a relatively safer O2O transfer in balanced replay (Lee et al., 2022), which further prioritizes the experience transitions that are closer to the current policy.

Unfortunately, all these methods require online estimation of distribution divergence or density ratio (for priority score or regularization weight). Excess conservatism can also slow down the online fine-tuning. A third category of methods avoid these complications by estimating the epistemic uncertainty of the $Q$-function, so that out-of-distribution actions carry a larger uncertainty which in turn yields conservative target values for Bellman backup (Jaksch et al., 2010; O'Donoghue et al., 2018; Osband et al., 2016; Kumar et al., 2019). However, it is generally hard to find calibrated uncertainty estimates, especially for deep neural nets (Fujimoto et al., 2019).

To resolve the aforementioned issues, we propose a novel alignment step for actor-critic RL that can be flexibly inserted between offline and online training, dispensing with any estimation of $Q$-function uncertainty, distribution divergence, or density ratio. Our key insight is drawn from soft actor-critic (SAC, Haarnoja et al., 2018), where the optimal entropy-regularized policy is simply the softmax of the $Q$-function. Now that the $Q$-function is generally problematic for out-of-distribution actions while the policy learned offline is assumed trustworthy (though still needs fine-tuning), it is natural to *align* the critic to the actor upon the completion of offline learning, so that the $Q$ function is tamed to be consistent with the policy under the softmax function, especially for those actions that lie outside the behavior policy. As a result, the online fine-tuning will only need to take the simple form of the standard SAC, and empirically the proposed method outperforms state-of-the-art fine-tuned robotic agents on various simulated tasks.

Our contributions and novelty can be summarized as follows:

- We propose a novel O2O RL approach that outperforms or matches the current SOTAs.
- Our approach does not rely on offline pessimism or conservatism, allowing it to transfer to a broader range of offline models.
- We propose, for the first time, discarding $Q$-values learned offline as a means to combat distribution shift in O2O RL. We also design a novel reconstruction of $Q$-functions for online fine-tuning.
- When offline data is not available at online fine-tuning – a very realistic scenario due to data privacy concerns, our method remains applicable and stable, while strong competitors such as balanced replay cease being applicable.

## 2  RELATED WORK

Decision transformer (Chen et al., 2021) and trajectory transformer (Janner et al., 2021) have recently been shown effective for offline reinforcement learning, where the batch trajectories' likelihood is maximized auto-regressively to model action sequences conditioned on a task. Zheng et al. (2022) extended them to *online* decision transformers (ODTs) by populating the replay buffer with online ODT rollouts labeled with hindsight experience replay. As a result, sequence modeling becomes effective for online fine-tuning. Our method remains in the actor-critic framework, and we demonstrate similar or superior empirical performance to ODT.

Behavior cloning often plays an important role in effective O2O RL. It can take the form of constraining the policy around the behavior policy under certain probability discrepancy measure, or simply imposing least square or cross-entropy regularizer to drive the policy to imitate transitions (Zhao et al., 2021; Fujimoto & Gu, 2021). Such a regularizer often requires delicate annealing, and to this end, Zhao et al. (2021) designed heuristic rules based on reward feedback. Recently, Kostrikov et al. (2022) employ behavior cloning to guide the extraction of policy from an expectile-based implicit $Q$-learning.

It is noteworthy that behavior cloning is also commonly used in imitation learning, where the goal is to imitate instead of outperforming the demonstrator, differing from the O2O setting. A number of efforts have been made to fuse it with RL for improvement (Lu et al., 2021). A similar line of research is to boost online learning from demonstration, (e.g., Hester et al., 2018; Reddy et al., 2019). However, they focus on accelerating online learning by utilizing offline data, and are not concerned about the safety or performance drop in porting the pre-trained policy to online.

## 3 PRELIMINARY

We follow the standard protocol that formulates a RL environment as a Markov decision process (MDP). An MDP $\mathcal{M}$ is often described as a 5-tuple $(\mathcal{S}, \mathcal{A}, \mathbb{P}, r, \gamma)$, where $\mathcal{S}$ is the state-space, $\mathcal{A}$ is the action space, $\mathbb{P} : \mathcal{S} \times \mathcal{A} \to \Delta(\mathcal{S})$ is the transition function, $R : \mathcal{S} \times \mathcal{A} \to \mathbb{R}$ is the reward function, and $\gamma \in [0, 1)$ is a discount factor. A policy is a distribution $\pi(a|s) \in \Delta(\mathcal{A})$, and the agent aims to find a policy that maximizes the expected return $\mathbb{E}_\pi[\sum_{t=0}^\infty \gamma^t r_t]$.

**Soft actor-critic**  To learn from offline data generated by a behavior policy, we will focus on off-policy RL methods. In particular, the soft actor-critic method (SAC, Haarnoja et al., 2017; 2018) learns a $Q$-function $Q_\mu(s, a)$ with parameter $\mu$, and a Gaussian policy $\pi_\theta(a|s)$ whose sufficient statistics are determined by a neural network with parameter $\theta$. Let $\mathbf{d}$ be the empirical distribution corresponding to the replay buffer, and we intentionally left it flexible on state, state-action, or transition. Then SAC alternates between updating the critic and actor by minimizing the following respective objectives:

$$\mathcal{L}_\pi^{\text{SAC}}(\theta, \mathbf{d}) := \mathbb{E}_{s \sim \mathbf{d}} \mathbb{E}_{a \sim \pi_\theta(\cdot|s)} \left[ \alpha \log \pi_\theta(a|s) - Q_\mu(s, a) \right], \tag{1}$$

$$\mathcal{L}_Q^{\text{SAC}}(\mu, \mathbf{d}) := \mathbb{E}_{(s,a,r,s',d) \sim \mathbf{d}} \left[ \left( Q_\mu(s, a) - y(r, s', d) \right)^2 \right], \tag{2}$$

$$\text{where} \quad y(r, s', d) := r + \gamma(1 - d) \mathbb{E}_{a' \sim \pi_\theta(\cdot|s')} \left[ Q_{\bar{\mu}}(s', a') - \alpha \log \pi_\theta(a'|s') \right]. \tag{3}$$

Here, $\alpha > 0$ is the temperature parameter, and $\bar{\mu}$ is the delayed $Q$-function parameter. If $\pi_\theta$ is based on a universal neural network, its optimal value that minimizes $\mathcal{L}_\pi^{\text{SAC}}(\theta, \mathbf{d})$ admits a closed form:

$$\pi_\theta(a|s) = \exp\left(\tfrac{1}{\alpha} Q_\mu(s, a)\right) \bigg/ \sum_{a \in \mathcal{A}} \exp\left(\tfrac{1}{\alpha} Q_\mu(s, a)\right). \tag{4}$$

In practice, one simply performs gradient descent steps on $\mathcal{L}_\pi^{\text{SAC}}$ because even if the network is universal, the value of $\theta$ that corresponds to the optimal solution (4) is hard to find.

It is important to note that adding a baseline function $Z(s)$ to $Q_\mu(s, a)$ does not change the optimal $\pi_\theta$ in (4), as long as $Z(s)$ does not depend on $a$. Therefore, given $\pi_\theta$, $Q_\mu(s, a)$ can be inferred as

$$Q_\mu(s, a) = Z(s) + \alpha \log \pi_\theta(a|s), \tag{5}$$

where $Z(s)$ provides additional freedom to fit other aspects of the problem; see Section 4.2.

## 4 ALIGNING CRITICS WITH ACTORS FOR OFFLINE-TO-ONLINE RL

We now detail our method that consists of three phases: offline, actor-critic (AC) alignment, and online. The whole procedure is summarized in Table 5 in Appendix A.

### 4.1 OFFLINE TRAINING

Motivating our offline training is TD3+BC (Fujimoto & Gu, 2021), which runs TD3 (Fujimoto et al., 2018) on the offline dataset with a behaviour cloning (BC) regularization (Bain & Sammut, 1995). Similar approaches such as SAC+BC can be found in Nair et al. (2020). However, we replaced TD3 with SAC to enable stochastic policies and to be consistent with the subsequent AC alignment, where the $Q$-function is obtained in closed-form under maximum entropy RL. We also replaced the BC regularization with a maximum likelihood (ML) regularizer, in order to be consistent with the online phase that also uses an ML regularizer (see Section 4.3). As a result, we naturally name our offline method as SAC+ML. We will compare our SAC+ML against TD3+BC in Appendix C.

**Actor update.** Let $\mathbf{d}$ be the empirical distribution of a mini-batch sampled from the offline dataset $\mathcal{D}$. The actor update of TD3+BC and SAC+ML aims to minimize the following respective objectives:

$$\mathcal{L}_\pi^{\text{TD3+BC}}(\theta, \mathbf{d}) = \mathbb{E}_{(s,a) \sim \mathbf{d}} \mathbb{E}_{b \sim \pi_\theta(\cdot|s)} \left[ -\lambda Q_\mu(s, b) + (b - a)^2 \right], \tag{6}$$

$$\mathcal{L}_\pi^{\text{SAC+ML}}(\theta, \mathbf{d}) = \mathbb{E}_{(s,a) \sim \mathbf{d}} \mathbb{E}_{b \sim \pi_\theta(\cdot|s)} \left[ -\lambda \left( Q_\mu(s, b) - \alpha \log \pi_\theta(b|s) \right) - \log \pi_\theta(a|s) \right], \tag{7}$$

where the hyperparameter $\lambda$ balances $Q$ values with the BC/ML regularization. In practice, we employed the clipped double $Q$-learning technique (Hasselt, 2010) to train two $Q$-networks $Q_{\mu_1}$ and $Q_{\mu_2}$. It is beneficial for both offline and online training (Fujimoto et al., 2018). $\lambda$ is then set to

$$\lambda := \omega \,/\, \mathbb{E}_{(s,a)\sim \mathbf{d}}[Q_\mu(s,a)], \quad \text{where} \quad Q_\mu := \min\{Q_{\mu_1}, Q_{\mu_2}\}, \tag{8}$$

and $\omega$ is a predetermined hyper-parameter. So $\lambda$ is recomputed after every critic update, requiring almost no additional computation.

**Critic update.** SAC+ML follows the same critic update as SAC in (2), except for the double $Q$ part:

$$\mathcal{L}_Q^{\text{SAC+ML}}(\mu_i, \mathbf{d}) := \mathbb{E}_{(s,a,r,s',d)\sim\mathbf{d}} \left[ (Q_{\mu_i}(s,a) - y(r,s',d))^2 \right] \tag{9}$$

$$\text{where} \quad y(r,s',d) := r + \gamma(1-d)\mathbb{E}_{a'\sim\pi_\theta(\cdot|s')}(Q_{\bar{\mu}}(s',a') - \alpha \log \pi_\theta(a'|s')), \tag{10}$$

and $\bar{\mu}$ is a delayed version of $\mu$, a.k.a. target network, with $Q_{\bar{\mu}}(s,a) = \min_{i\in\{1,2\}} Q_{\bar{\mu}_i}(s,a)$, akin to $Q_\mu$. The temperature update tries to reduce the following objective over $\alpha > 0$:

$$\mathcal{L}_{\text{temp}}^{\text{SAC+ML}}(\alpha, \mathbf{d}) := \mathbb{E}_{s\sim\mathbf{d}}\mathbb{E}_{a\sim\pi_\theta(\cdot|s)} \left[ -\alpha \left( \log \pi_\theta(a|s) - \bar{\mathcal{H}} \right) \right]. \tag{11}$$

Here, $\bar{\mathcal{H}} > 0$ is the target entropy value, a hyper-parameter specified a priori. The Lagrange multiplier $\alpha$ is automatically tuned in (11), enforcing the upper bound of the entropy of $\pi_\theta$ by $\bar{\mathcal{H}}$. The pseudo-code of SAC+ML is relegated to Appendix A.

## 4.2 ACTOR-CRITIC ALIGNMENT

At the end of offline learning, the learned policy $\pi_{\theta_0}$ often performs reasonably well, and is ready for online fine-tuning. So we denote the policy with index 0. In conventional actor-critic, the critic is supposed to be updated frequently enough to accurately pursue the state-action values for the current policy. However, even if such updates are conducted proactively, the distribution shift problem in O2O still plagues the critic under deep net approximation, because the $Q$-values are not trustworthy beyond what has been visited under the behavior policy. So the over-estimated $Q$-values can rapidly destroy the learned actor and critic through Bellman backup.

In order to avoid this issue, we propose taming the out-of-distribution $Q$-values by directly aligning the critics with the actors, as a post-processing step for offline learning, or an initialization step for online learning. In particular, inspired by (5), we choose to discard the $Q_{\mu_i}$ learned from the offline phase, and reset them into[1]

$$Q_i(s,a) = \log \pi_{\theta_0}(a|s) + Z_{\psi_i}(s). \tag{12}$$

The baseline $Z_{\psi_i}(s)$ can be naturally calibrated by minimizing the Bellman residual on offline data:

$$\mathcal{L}_Z^{\text{SAC+ML}}(\psi_i, \mathbf{d}) := \mathbb{E}_{(s,a,r,s',d)\sim\mathbf{d}} \left[ (\log \pi_{\theta_0}(a|s) + Z_{\psi_i}(s) - y(r,s',d))^2 \right] \tag{13}$$

$$\text{where} \quad y(r,s',d) := r + \gamma(1-d)\mathbb{E}_{a'\sim\pi_{\theta_0}(\cdot|s')} \left[ \log \pi_{\theta_0}(a'|s') + Z_\psi(s') \right], \tag{14}$$

$$Z_\psi := \min\{Z_{\psi_1}, Z_{\psi_2}\}. \tag{15}$$

Here, $Z_\psi$ in (14) employs a standard semi-gradient. This optimization is simply a regression problem and can be conducted by Adam (Kingma & Ba, 2015). The details are deferred to Appendix A.1, where pseudo-code is also given in Algorithm 1.

**Generality.** Thanks to this alignment step that disregards the $Q$-function learned offline, the offline learning algorithm is not limited to SAC+ML, even though it is more favorable. In Section 6.3, we will show that our alignment approach can be well applied to the offline policy learned from CQL.

## 4.3 ONLINE TRAINING

During the online fine-tuning, we restore the full flexibility of $Q$-functions by using the following parameterization:

$$Q_{\phi_i}(s,a) := \log \pi_{\theta_0}(a|s) + R_{\phi_i}(s,a), \quad \text{where} \quad R_{\phi_i}(s,a) \text{ is } \textbf{initialized} \text{ with } Z_{\psi_i}(s). \tag{16}$$

---

[1]Compared with (5), it appears that we have set $\alpha$ there to 1, while its value at the end of offline learning is rarely close to 1. This creates no contradiction, however, because $\log \pi_{\theta_0}$ will be used to parameterize the online $Q$-function as in (16), and the $\alpha$ for online phase SAC is initialized to 1. So their product, passed through the softmax, will recover $\pi_{\theta_0}$.

Such an initialization can be simply implemented by loading the weights of $Z_{\psi_i}$ and setting the weights corresponding to action to zeros. It is noteworthy that one should refrain from constraining $Q$ to closed-form manifold induced by the latest $\pi_\theta$ throughout the online phase, i.e., setting $Q_{\phi_i}(s, a)$ to $\log \pi_\theta(a|s) + Z_{\phi_i}(s)$ for some trainable baseline $Z_{\phi_i}$. This is because it would lead to no improvement of the policy. As such, we only leverage the closed-form for initialization.

The update on temperature is exactly the same as (11), and the update on critic resembles that of the offline phase in (9), except that the training variable is now only $R_{\phi_i}(s, a)$. In particular, we adapt SAC critic update to our $Q_\phi$, along with standard tricks of target network and double $Q$-clipping:

$$\mathcal{L}_Q(\phi_i, \mathbf{d}) := \mathbb{E}_{(s,a,r,s',d)\sim\mathbf{d}}\left[\left(\log \pi_{\theta_0}(a|s) + R_{\phi_i}(s, a) - y(r, s', d)\right)^2\right], \tag{17}$$

where $y(r, s', d) := r + \gamma(1-d)\mathbb{E}_{a'\sim\pi_\theta(\cdot|s')}\left[\log \pi_{\theta_0}(a'|s') + R_{\bar{\phi}}(s', a') - \alpha \log \pi_\theta(a'|s')\right]$. (18)

The actor's objective follows from the **vanilla SAC**, and can be written as follows with $\mathbf{d}$ being a mini-batch sampled from the replay buffer, $R_\phi := \min_{i\in\{1,2\}} R_{\phi_i}$, and $Q_\phi := \log \pi_{\theta_0} + R_\phi$:

$$\mathcal{L}_\pi(\theta, \mathbf{d}) := -\mathbb{E}_{s\sim\mathbf{d}}\mathbb{E}_{a\sim\pi_\theta(\cdot|s)}\left[Q_\phi(s, a) - \alpha \log \pi_\theta(a|s)\right], \tag{19}$$

$$= -\mathbb{E}_{s\sim\mathbf{d}}\mathbb{E}_{a\sim\pi_\theta(\cdot|s)}\left[R_\phi(s, a) - \alpha \log \pi_\theta(a|s)\right] - \underbrace{\mathbb{E}_{s\sim\mathbf{d}}\mathbb{E}_{a\sim\pi_\theta(\cdot|s)}\left[\log \pi_{\theta_0}(a|s)\right]}_{\text{penalizing deviation of } \pi_\theta \text{ from } \pi_{\theta_0}}. \tag{20}$$

**Behavior cloning in** (20). Naturally unfolding from SAC using the parameterization (16) is the expectation of log-likelihood of $\pi_{\theta_0}$ under $\pi_\theta$ in (20), a maximum-likelihood term that enforces the actions favored by the new policy $\pi_\theta$ to also enjoy a high log-likelihood under the offline policy $\pi_{\theta_0}$. Different from AdaBC (Zhao et al., 2021), we sidestepped an ad-hoc introduction of behavior cloning regularization and tweaking of its weight. This regularization is *not* applied on the offline data, but on the policy $\pi_{\theta_0}$ achieved by offline learning. To be consistent, we also used the maximum-likelihood regularization in the offline training.

One might argue that this interpretation is artificial because, after all, the $\log \pi_{\theta_0}$ term can be subsumed into the free variable $R_{\phi_i}$ in (16), obliterating this BC regularizer in (20). This in fact makes sense if the entire optimization is convex and the range of $R_{\phi_i}$ as a function set is closed under addition with $\log \pi_{\theta_0}$. However, since $R_{\phi_i}$ is a neural network, such conditions do not hold true. As a result, the composite form in (16) does play a crucial role in the good empirical performance, which is manifested in our ablation study in Section 6.5.

In practice, we also introduced two techniques to stabilize online learning. The first $\beta$-clipping trick addresses the excessively large magnitude of $\log \pi_{\theta_0}$ by capping its absolute values. The second critic interpolation gives the flexibility to balance between safety transfer and policy improvement. For the sake of space, they are deferred to Appendix A.2.

## 5 ILLUSTRATION OF ALIGNMENT UNDER DISTRIBUTION SHIFT

We first demonstrate how the critic alignment makes the $Q$-function more consistent with the real actions in the offline dataset, compared with the $Q$-values learned from offline actor-critic. We trained SAC+ML on the halfcheetah-medium dataset, and sampled in-distribution states from it. To sample out-of-distribution states, we resorted to the halfcheetah-expert dataset, and the details are available in Appendix D. Figure 5 there further illustrates the difference of these two state distributions.

Figure 1 (top row) compares the $Q$-values learned from SAC+ML and our aligned/reconstructed $Q$-values, where the state-action pair is sampled in-distribution. The bottom row shows a similar comparison, but on out-of-distribution samples. The actions have 6 dimensions, and for the $i$-th subplot, we perturbed the $i$-th dimension in $[-1, 1]$, with all the other dimensions fixed. Clearly, the offline learned $Q$-values are often inconsistent with the real action from the dataset, even for in-distribution samples. But our alignment much improves the consistency, which encourages the policy to stay close to the offline policies, safeguarding the process of transfer.

## 6 EXPERIMENTS

We next compared our actor-critic alignment method (ACA) with a number of state-of-the-art methods as summarized in Table 1. Although CQL was not developed for O2O transfer, we still included

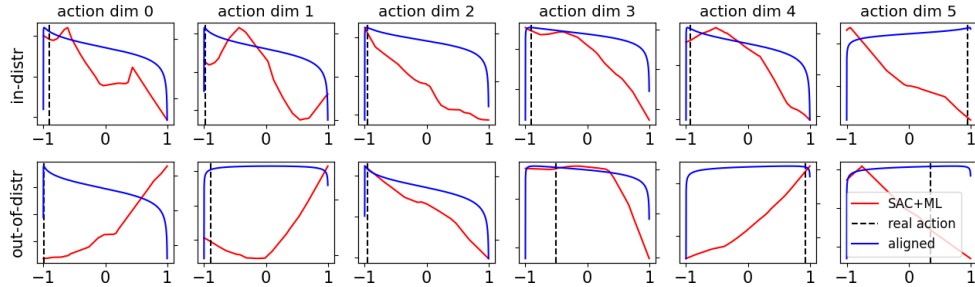

Figure 1: SAC+ML $Q$-values v.s. aligned $Q$-values for in-distribution sample (top row) and out-of-distribution sample (bottom row). Left $y$-axis: SAC+ML $Q$-values, right $y$-axis: aligned $Q$-values. Since only the trend of each curve matters, we omit the $y$-axis tick values.

Table 1: Baseline algorithms for O2O RL. See acronyms below.

| Flag | | Name | Offline | Online | Description |
|---|---|---|---|---|---|
| | → | SAC→ACA (Ours) | SAC+ML | Algorithm 3 | Our method init from SAC+ML |
| | → | SAC→SAC | SAC+ML | SAC | SAC init from SAC+ML |
| | → | CQL→BR | CQL | SAC w/ BR | Balanced replay init from CQL |
| | → | CQL→SAC | CQL | SAC | SAC init from CQL |
| | → | AWAC | AWAC | AWAC | AWAC init from AWAC |
| | | SAC | - | SAC | SAC from scratch |

it due to its strong performance. Implementation of our ACA algorithm can be found anonymously at Online Supplementary, along with the pre-trained models.

Our experiments aim to demonstrate:

- SAC→ACA matches or outperforms SOTAs, e.g., balanced replay (BR, Lee et al., 2022), advantage weighted actor critic (AWAC, Nair et al., 2020), and online decision transformer (ODT, Zheng et al., 2022);
- Direct transfer such as SAC→SAC and CQL→SAC suffers significant performance drop;
- Transfer from offline method significantly outperforms training SAC online from scratch. We will additionally present ablation studies to examine various components of ACA.

### 6.1 COMPARISON WITH BASELINE METHODS

We used three environments from the datasets D4RL-v2 (Fu et al., 2020), including HalfCheetah, Hopper, and Walker2d. Each of them has five levels. All offline/online experiments ran 5 random seeds. We ran all offline algorithms for 500 episodes with 1000 mini-batches each, and all online experiments for 100 episodes with 1000 environment interactions each. This protocol is quite commonly used. More implementation details are deferred to Appendix B.

Figure 2 shows the average return as a function of training episodes, achieved at each offline model (left half of the subplots) and online model (right half). Since SAC→ACA and SAC→SAC share the same offline method, their curves coincide on the left of the subplots, with the green curve shown only (no blue) on the left. A similar situation occurs to CQL→BR and CQL→SAC, and only the purple curve is shown on the left half (no pink).

In Figure 2, CQL→SAC (purple→pink) drops significantly on the expert level (fifth row) and medium-expert level (fourth row). SAC→SAC (green→blue) drops in almost all cases, except random (first row) and medium-replay (third row). It is clear that our SAC→ACA (green) barely suffers performance drop. The only exception is Hopper-medium-expert, but all other methods (except AWAC which performs poorly offline) also suffer a drop there, while ours recover most rapidly. Besides, ours offers comparable policy improvement to the strongest baseline, which is CQL→BR in most cases.

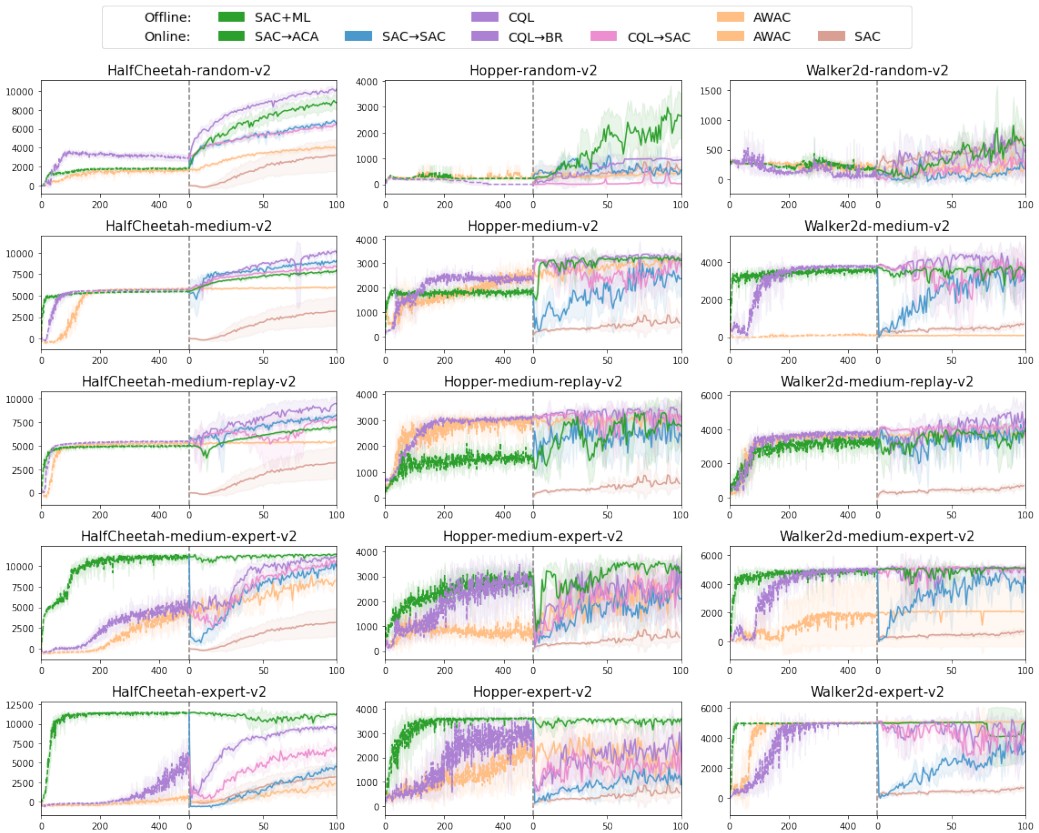

Figure 2: Comparing SAC→ACA (ours) with other baselines for offline-to-online RL. The shaded areas stand for the standard deviation. Refer to Table 1 for legend meanings.

Table 2: Average normalized D4RL scores of various O2O methods. Outside parenthesis: scores at the end of 100k online steps. Inside parenthesis: the increase of that score upon the end of offline training. HC = HalfCheetah, H = Hopper, W = Walker2d.

| Dataset | Env | SAC→SAC | CQL→SAC | Score(δ) AWAC | CQL→BR | SAC→ACA (ours) |
|---|---|---|---|---|---|---|
| Random | HC | 54.60(37.31) | 54.00(28.70) | 34.26(18.94) | 84.36(59.06) | 72.60(55.31) |
| | H | 17.62(9.69) | 1.37(0.72) | 16.85(3.03) | 29.80(29.15) | 81.85(73.92) |
| | W | 3.86(0.50) | 3.91(3.25) | 4.15(-0.58) | 10.05(9.39) | 12.42(9.06) |
| Medium | HC | 75.20(28.86) | 69.52(21.10) | 50.48(1.54) | 82.95(34.52) | 66.58(20.25) |
| | H | 73.39(19.08) | 89.77(16.30) | 97.53(24.48) | 98.14(24.67) | 96.54(42.24) |
| | W | 79.63(-1.53) | 81.78(-0.70) | 1.93(-0.54) | 76.36(-6.11) | 74.66(-6.50) |
| Med.-Replay | HC | 68.90(26.37) | 63.91(18.01) | 46.84(2.42) | 78.36(32.46) | 59.03(16.50) |
| | H | 74.04(25.22) | 92.01(-3.95) | 95.98(0.00) | 97.25(1.28) | 85.54(36.72) |
| | W | 85.40(23.21) | 79.28(0.89) | 80.81(2.97) | 100.06(21.68) | 85.17(22.98) |
| Med.-Expert | HC | 82.15(-11.38) | 87.85(46.77) | 68.75(32.30) | 91.80(50.73) | 93.74(0.21) |
| | H | 65.44(-27.64) | 80.46(-13.26) | 73.13(47.50) | 78.51(-15.22) | 98.02(4.94) |
| | W | 87.18(-20.95) | 107.03(-2.63) | 45.21(4.45) | 104.43(-5.22) | 110.54(2.42) |
| Expert | HC | 38.17(-55.42) | 55.39(6.86) | 21.23(14.83) | 79.69(31.16) | 93.14(-0.46) |
| | H | 28.20(-82.68) | 67.88(-15.48) | 57.97(-12.85) | 68.55(-14.81) | 110.21(-0.67) |
| | W | 67.76(-40.45) | 81.92(-27.24) | 110.68(0.80) | 110.65(1.50) | 109.59(1.38) |
| Total | | 901.54(-69.80) | 1016.08(79.35) | 805.80(139.29) | 1190.96(254.23) | 1249.65(278.31) |

Since different baselines in Table 1 employ different offline methods, it is not reasonable to compare fine-tuning methods based only on their online performance. Therefore, we provided in Table 2 the increase of return achieved online compared with the final offline policy. As the numbers in the parenthesis there show, SAC→ACA attains the highest improvement (as well as the final score), and BR appears the best among all other baseline methods.

## 6.2 COMPARISON WITH ONLINE DECISION TRANSFORMER

Since ODT is not based on dynamic programming, we compared it with SAC→ACA in this separate section. As Zheng et al. (2022) experimented using 200k online samples and averaged over 10 seeds, we ran SAC+ML with 5 additional seeds and ran 200k online steps for all 10 SAC+ML runs, to make the comparison fair.

Table 3: Comparing SAC→ACA with online decision transformer (ODT), with a focus on the online improvement upon offline policy ($\delta_{\text{ODT}}$ and $\delta_{\text{ACA}}$).

| Dataset | Environment | ODT(offline) | ODT(200k) | $\delta_{\text{ODT}}$ | SAC+ML | ACA(200k) | $\delta_{\text{ACA}}$ |
|---|---|---|---|---|---|---|---|
| Medium | HalfCheetah | $42.72_{\pm0.46}$ | $42.16_{\pm1.48}$ | -0.56 | $46.40_{\pm0.30}$ | $72.67_{\pm3.01}$ | 26.28 |
| | Hopper | $66.95_{\pm3.26}$ | $97.54_{\pm2.10}$ | 30.59 | $56.93_{\pm4.12}$ | $99.32_{\pm7.82}$ | 42.39 |
| | Walker2d | $72.19_{\pm6.49}$ | $76.79_{\pm2.30}$ | 4.60 | $79.36_{\pm2.25}$ | $76.05_{\pm20.57}$ | -3.30 |
| Med.-Replay | HalfCheetah | $39.99_{\pm0.68}$ | $40.42_{\pm1.61}$ | 0.43 | $42.18_{\pm0.53}$ | $64.29_{\pm2.97}$ | 22.11 |
| | Hopper | $86.64_{\pm5.41}$ | $88.89_{\pm6.33}$ | 2.25 | $49.25_{\pm6.08}$ | $103.17_{\pm3.08}$ | 53.92 |
| | Walker2d | $68.92_{\pm4.79}$ | $76.86_{\pm4.04}$ | 7.94 | $63.20_{\pm10.12}$ | $82.09_{\pm27.66}$ | 18.89 |
| Total (w/o hopper-mr) | | 290.77 | 333.77 | 43.00 | 288.06 | 394.41 | 106.35 |
| Total (all) | | 377.41 | 422.66 | 45.25 | 337.31 | 497.58 | 160.27 |

As shown in Table 3, for almost all medium and medium-replay tasks, our SAC→ACA outperforms ODT in both final performance and performance increase ($\delta$). We also note that ODT(offline) outperforms SAC+ML in the hopper-medium-replay task by a large margin, which leaves our approach more room to improve. Therefore, we made the same comparison by excluding the hopper-medium-replay task. In this case, ODT and ours were initialized from roughly the same performance, and ours still outperforms ODT in both total final performance and total performance increase.

## 6.3 FLEXIBILITY IN INITIALIZATION FROM DIFFERENT OFFLINE METHODS

A key advantage of our alignment method lies in the flexibility of leveraging any offline RL method, as long as it outputs a parameterized Gaussian policy, because the $Q$-function is reset anyway. In contrast, SOTA methods sometimes require certain properties in the offline method such as pessimism. For example, BR's performance depends critically on the use of CQL.

To demonstrate our flexibility, we adopted CQL for offline learning and made a simple change to the alignment step which, in (13), clips $\log \pi_{\theta_0}$ to 0 when it is negative. In comparison, we also tested BR by using SAC+ML as the offline learner. Figure 3 shows the results of ACA/BR initialized from SAC+ML/CQL. While the performance of our approach does not change much when initializing from different offline models, BR shows significant performance drops when it is initialized from SAC+ML, i.e. non-pessimistic offline training.

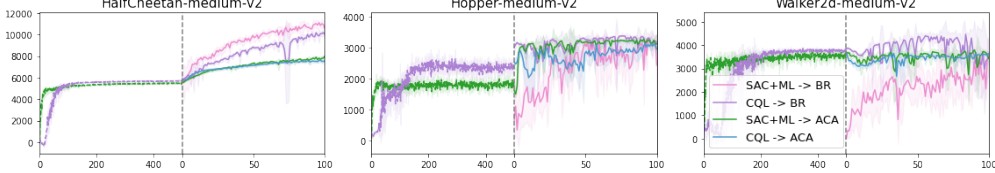

Figure 3: ACA and BR initialized from different offline methods. ACA could achieve similar performance while initializing from both SAC+ML/CQL. BR requires CQL initialization.

## 6.4 ONLINE TRAINING WITHOUT OFFLINE DATA

When the application precludes the accessibility of offline data during online fine-tuning, we re-ran the benchmarks for medium-replay, medium-expert, expert. There is obviously no reason to replay offline data at random level, and empirically, we observed that online fine-tuning already performed well on medium when no offline data was replayed.

Figure 6 in Appendix F shows the online average return of our method without using offline data, compared with other baselines which also do not access offline data during online fine-tuning. The balance replay algorithm requires offline data. So compared with Figure 2, we no longer have the purple line that corresponds to CQL→BR. It turns out that all the other baselines retain similar online performance as in Figure 2, which had been shown inferior to our SAC→ACA. Table 4 further highlights that SAC→ACA does not exhibit significant change in online performance in the absence of offline data.

Table 4: Scores for SAC→ACA at 100k online steps, and its increase from offline result (in parenthesis). Comparison is made between with or without offline data. HC = HalfCheetah, H = Hopper, W = Walker2d.

| Dataset | Env | Score($\delta$) | |
|---|---|---|---|
| | | w/ offline data | w/o offline data |
| Med.-Replay | HC | 59.03(16.50) | 59.48(16.95) |
| | H | 85.54(36.72) | 77.19(28.37) |
| | W | 85.17(22.98) | 84.27(22.08) |
| Med.-Expert | HC | 93.74(0.21) | 93.81(0.28) |
| | H | 98.02(4.94) | 105.67(12.59) |
| | W | 110.54(2.42) | 110.93(2.81) |
| Expert | HC | 93.14(-0.46) | 90.76(-2.83) |
| | H | 110.21(-0.67) | 109.22(-1.66) |
| | W | 109.59(1.38) | 110.52(2.31) |
| Total | | 844.98(84.02) | 841.84(80.88) |

## 6.5 ABLATION STUDY

As mentioned in Section 4.3, $\log \pi_{\theta_0}$ in (20) can be considered as a "behaviour cloning" regularization. One may wonder whether this, instead of actor-critic alignment, is the primary contributor to the empirical effectiveness. We therefore conducted the following ablation study, which answers this question in the negative.

In contrast to the parameterization of online $Q$-function in (16), we designed two alternatives. The first directly copies the $Q$-function from the conclusion of offline learning, and then fine-tunes it online. The second adopts the same decomposed parameterization as in (16), but initializes $R_{\phi_i}$ with the offline learned $Q_{\mu_i}$, instead of $Z_{\psi_i}$. As a result, $\log \pi_{\theta_0} + R_\phi$, whose expectation serves in the actor objective, becomes a regular offline-trained $Q$ function with a regularizer $\log \pi_{\theta_0}$, instead of an actor-critic alignment. The update objectives of the two ablations are relegated to Appendix E.

Figure 4 shows that on tasks vulnerable to transfer risk such as hopper-medium and walker-medium (second and third subplots), the two ablation alternatives suffer clear performance drop due to the attributed error in the offline trained $Q$-functions. However, some tasks can be less vulnerable. For example, on halfcheetah-medium, Figure 2 shows that SAC→SAC (green→blue) only suffers a small amount of drop, although it employs no mechanism to combat distribution shift. In such a task, the two ablation alternatives remain competitive to no surprise (first subplot).

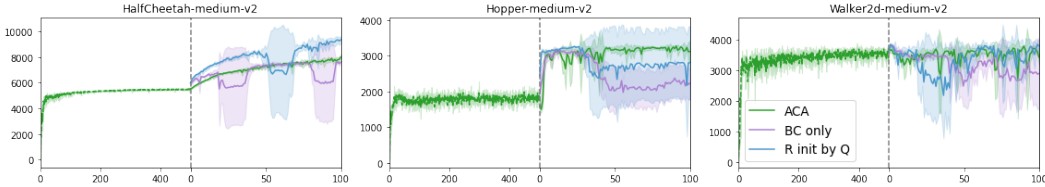

Figure 4: Ablation study: We keep the actor update same as Eq (20), we however change the critic update (details found in Appendix 6.5) to see show that the re-parameterized $Q$-function is critical. **BC only** (ablation 1): $R_\phi$ is seen as the online critic and is initialized by $Q_\mu$. Critic updates are made as regular SAC critic update without our re-parameterization. It therefore can be seen as SAC with behavior cloning regularized actor update. **$R_\phi$ init by $Q_\mu$** (ablation 2): We keep the ACA framework but initialize $R_\phi$ by $Q_\mu$ instead of $Z_\psi$ to show the importance of the baseline.

## 7 CONCLUSION AND FUTURE WORK

We proposed a new actor-critic alignment method that allows safe offline-to-online reinforcement learning and achieves strong empirical performance. To combat distribution shift, we designed a novel approach that disregards offline learned $Q$-functions, and reconstructs it based on the learned policy using a closed-form that is motivated from the entropy-regularized actor update. Since it does not need an offline critic, online actor-critic fine-tuning is made possible for offline learned decision transformer, as well as other supervised learning methods such as RvS (Emmons et al., 2022).

**Reproducibility Statement.** Our ACA implementation can be found anonymously at Online Supplementary, along with the pre-trained offline models. More implementation details regarding offline approaches and baselines are available in Appendix B, and our choice of hyper-parameters can be found in Appendix I.

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

# A  ALGORITHM DETAILS

The pseudo-code of the offline, alignment, and online phases is provided in Algorithm 1, 2, and 3, respectively.

Table 5: Learning in three phases for offline-to-online RL with actor-critic alignment

| Phase | Learnable Component | Description |
|---|---|---|
| Offline training | $\pi_\theta(a\|s), Q_\mu(s,a), \alpha$ | Run offline RL to obtain offline policy $\pi_\theta(a\|s)$ and $Q$-function $Q_\mu(s,a)$. |
| Actor-critic alignment | $Z_\psi(s)$ | Fit the baseline $Z_\psi(s)$ by minimizing the Bellman residue. |
| Online training | $\pi_\theta(a\|s), R_\phi(s,a), \alpha$ | Run actor-critic on policy $\pi_\theta(a\|s)$ and critic $Q_\phi(s,a) \coloneqq \log \pi_{\theta_0}(a\|s) + R_\phi(s,a)$, where $R_\phi(s,a)$ is initialized by $Z_\psi(s)$. |

---

**Algorithm 1** Offline SAC+ML

Initialize parameters $\theta, \alpha, \psi_i, \mu_i, \bar{\mu}_i$ for $i \in \{1,2\}$
**for** each iteration **do**
    sample mini-batch from dataset $\mathcal{D}$
    update $\alpha$ with Eq. (11)
    update $\mu_i$ with Eq. (9) for $i \in \{1,2\}$
    update $\theta$ with Eq. (7)
    update $\psi_i$ with Eq. (13) for $i \in \{1,2\}$
    $\bar{\mu}_i \leftarrow \tau\mu_i + (1-\tau)\bar{\mu}_i$ for $i \in \{1,2\}$
**end for**

---

**Algorithm 2** Actor-critic Alignment

**Require:** $\theta, \alpha, \psi_i, \mu_i, \bar{\mu}_i$, for $i \in \{1,2\}$
    Initialize parameters $\phi_i, \bar{\phi}_i$, for $i \in \{1,2\}$
    Set $R_{\phi_i}(s,a) \leftarrow Z_{\psi_i}(s)$, for $i \in \{1,2\}$
    Copy $\bar{\phi}_i \leftarrow \phi_i$
    Copy $\theta_0 \leftarrow \theta$
    Reset $\alpha \leftarrow 1$
    Delete $\mu_i$ and $\bar{\mu}_i$, for $i \in \{1,2\}$

---

## A.1  OPTIMIZATION OF $Z_{\psi_i}$

Since the alignment objective (13) needs to access offline data, we blended it into the offline training as shown in the second last step of Algorithm 1. It is noteworthy that this is only for the convenience of implementation, and the $\psi_i$ values do *not* have any influence on SAC+ML training itself. Conversely, the optimized value of $\psi_i$ provides a good initialization for a standalone optimization of objective (13). In practice, we observed that the $\psi_i$ found from offline training is good enough, and we just directly used them to initialize the online critic $R_{\phi_i}$.

## A.2  TECHNIQUES TO STABILIZE ONLINE LEARNING

We propose $\beta$-clipping trick and critic interpolation to achieve better empirical performance. As $\log \pi_{\theta_0}$ is unbounded below, the training can be numerically unstable, $\beta$-clipping trick bounds the term to stabilize training. And critic interpolation gives the flexibility to balance between safety transfer and policy improvement.

### A.2.1  $\beta$-CLIPPING TRICK

As in the course of online learning, the magnitude of $|\log \pi_{\theta_0}|$ can sometimes be an/several order larger than $R_\phi$, which leads to very instable critic training. Given $\pi_{\theta_0}(a|s)$ is parameterized by squashed Gaussian distribution $N(\mu_s, \sigma_s^2)$, we clip the $\log \pi_{\theta_0}$ term, as follows

$$\text{CLIP}_\beta(\log \pi_{\theta_0}(a|s)) \coloneqq \text{SoftPlus}\Big(\log \pi_{\theta_0}(a|s) - C_\beta(s)\Big) + C_\beta(s) \tag{21}$$

$$\text{where} \quad C_\beta(s) = \min\{\log \pi_{\theta_0}(\mu_s - \beta\sigma_s|s), \log \pi_{\theta_0}(\mu_s + \beta\sigma_s|s)\}. \tag{22}$$

---

**Algorithm 3** Online training

---

**Require:** $\theta, \theta_0, \alpha, \phi_i, \bar{\phi}_i$ from Algo. 2
  **if** $\mathcal{D}$ is accessible **then**
    Initialize replay buffer $\mathcal{B}$ with top $N$ trajectories
  **else**
    Initialize replay buffer $\mathcal{B} \leftarrow \varnothing$
  **end if**
  **for** each iteration **do**
    sample a mini-batch from buffer $\mathcal{B}$
    update $\alpha$ with Eq. (11)
    update $\phi_i$ with Eq. (30) for $i \in \{1, 2\}$
    update $\theta$ with Eq. (29)
    $\bar{\phi}_i \leftarrow \tau \phi_i + (1 - \tau)\bar{\phi}_i$
  **end for**

---

Here, $\beta$ is a hyper-parameter, and $\text{SoftPlus}(x) = \log(1 + \exp(x))$. Essentially it clips $\log \pi_{\theta_0}(a|s)$ at $C_\beta(s)$. This $\text{CLIP}_\beta(\cdot)$ operator bounds the $\log \pi_{\theta_0}$ term in a reasonable range, and also requires minimal tuning of hyper-parameter, see Section G and Section H for details. Using $\text{CLIP}_\beta$, we define $Q_\phi^\beta$ as

$$Q_\phi^\beta(s, a) := \text{CLIP}_\beta(\log \pi_{\theta_0}(a|s)) + R_\phi(s, a). \tag{23}$$

Now, the clipped online actor/critic updates can be summarized by

$$\mathcal{L}_\pi^\beta(\theta, \mathbf{d}) = \mathbb{E}_{s \sim \mathbf{d}} \mathbb{E}_{a \sim \pi_\theta} \left[ \alpha \log \pi_\theta(a|s) - Q_\phi^\beta(s, a) \right], \tag{24}$$

$$\mathcal{L}_Q^\beta(\phi_i, \mathbf{d}) = \mathbb{E}_{(s,a,r,s',d) \sim \mathbf{d}} \left[ \left( Q_{\phi_i}^\beta(s, a) - y(r, s', d) \right)^2 \right], \tag{25}$$

$$y(r, s', d) = r + \gamma(1 - d)\mathbb{E}_{a' \sim \pi_\theta(\cdot|s')} \left[ Q_\phi^\beta(s', a') - \alpha \log \pi_\theta(a'|s') \right]. \tag{26}$$

### A.2.2 CRITIC INTERPOLATION

At the initial phase of online training, $\text{CLIP}_\beta(\log \pi_{\theta_0}(a|s))$ dominates the actor update, safeguarding the policy. As training proceeds, $R_\phi$ grows to overcome the barrier and starts to improve the policy. Ideally, we wish to finely control such a junction so that the safety of O2O transition does not excessively slow down the policy improvement. To this end, we introduce an interpolation between closed-form initialized critic and restriction-free critic. We call it critic interpolation, which can be written as

$$Q_\phi^{k,\beta}(s, a) := k \Big( \underbrace{\text{CLIP}_\beta\big(\log \pi_{\theta_0}(a|s)\big) + R_\phi(s, a)}_{\text{closed-from initialized critic}} \Big) + (1 - k) \underbrace{R_\phi(s, a)}_{\text{restriction-free critic}} \tag{27}$$

$$= k \times \text{CLIP}_\beta\big(\log \pi_{\theta_0}(a|s)\big) + R_\phi(s, a). \tag{28}$$

We set $k = 1$ at $t = 0$ to assert closed-form initialization. Then we linearly decay $k$ during the course of online training, allowing a transition from closed-form initialization to free SAC update. The detailed decaying rate can be found in Appendix I.

## A.3 CONCLUDED ONLINE TRAINING

Our final online update rules are summarized as follows:

$$\mathcal{L}_\pi^{\text{online}}(\theta, \mathbf{d}) = \mathbb{E}_{s \sim \mathbf{d}} \mathbb{E}_{a \sim \pi_\theta} \left[ \alpha \log \pi_\theta(a|s) - Q_\phi^{k,\beta}(s, a) \right] \tag{29}$$

$$\mathcal{L}_Q^{\text{online}}(\phi_i, \mathbf{d}) = \mathbb{E}_{(s,a,r,s',d) \sim \mathbf{d}} \left[ \left( Q_\phi^{k,\beta}(s, a) - y(r, s', d) \right)^2 \right] \tag{30}$$

$$y(r, s', d) = r + \gamma(1 - d)\mathbb{E}_{a' \sim \pi_\theta(\cdot|s')} \left[ Q_\phi^{k,\beta}(s', a') - \alpha \log \pi_\theta(a'|s') \right]. \tag{31}$$

# B    IMPLEMENTATIONS

Overall, all our implementations are from or based on d3rlpy (Takuma Seno, 2021), a popular RL library that specialized for offline RL. Using the same lib helps us to minimize the impact of implementation difference. Many of our baselines (see Table 9) are implemented upon SAC, with changes proposed in their original papers, respectively.

## B.1    GENERAL IMPLEMENTATION DETAILS

**Evaluation protocol:** All offline/online experiments ran 5 random seeds. We ran all offline algorithms for 500 episodes with 1000 mini-batches each, and all online experiments for 100 episodes with 1000 environment interactions each. After each episode, we conducted 10 evaluations and computed the average return. Results reported are mean and std of average returns, over 5 random seeds.

**Choice of offline checkpoints:** Evaluating in the offline phase, in fact, requires online interactions. Therefore we do **not** pick the best-performed checkpoints. Instead, we use the last checkpoints as our initialization models, for online.

**Squashed Gaussian:** For methods with stochastic policies, we parameterized their policies by unimodal Gaussian, and applied the squashed Gaussian trick (Haarnoja et al., 2018) to bound the range of action to $[-1, 1]$.

**Buffer initialization:** We followed the instructions in AWAC and BR papers on initializing online replay buffers. For AWAC, we added all transitions in $\mathcal{D}$ to the buffer $\mathcal{B}$. And for BR, we refer to their original implementation at this URL for details. For SAC→SAC and CQL→SAC, we added all transitions in $\mathcal{D}$ to the buffer $\mathcal{B}$ as well, as there is no explicit instructions or common protocols. All replay buffer sizes were set to be 1e6, unless specified in the Appendix I.

## B.2    OFFLINE

**AWAC and CQL:** We used d3rlpy implementations for AWAC and CQL.

**SAC+ML:** Our SAC+ML implementation was adapted from d3rlpy's TD3+BC implementation, with changing the actor update rule to Eq. (7), and adding the learning of baseline $Z_\psi$.

## B.3    ONLINE

**Training details for online:** For all methods, we made a temperature (if applicable), a critic, and an actor update after every environmental interaction, if there were enough transitions (i.e. more than batch size) in the replay buffer. Target networks were all updated in a Polyak averaging fashion, where the step size $\tau = 0.005$ for all experiments. See Section I for more hyper-parameter details. And online results, reported in tables, were also using the last checkpoints instead of best-performed ones.

**SAC:** We used d3rlpy implementation for SAC.

**SAC→SAC and CQL→SAC:** We simply loaded offline-trained SAC+ML and CQL, respectively, and then ran SAC online.

**BR:** We adapted all parts that related to the prioritized replay from the official BR implementation, to a d3rlpy SAC implementation base, as the original BR paper also run SAC online.

**ACA (ours:)** Implementation of our approach can be found anonymously at Online Supplementary. In addition to Algorithm 3, we also did gradient norm clipping to actor updates, which is commonly used in RL implementations.

## C  SAC+ML VS. TD3+BC

We would like to emphasize that our goal is not to propose a stronger offline RL method. Table 6 is presented to show that our SAC+ML modification performs comparably to the original SOTA method, TD3+BC.

TD3+BC results in Table 6 were copied from appendix C.3 of their paper (Fujimoto & Gu, 2021). The evaluation protocol is identical to theirs: (1) all experiments were done in D4RL-v2 datasets; (2) and the results reported were from the last evaluation step, averaged over 5 random seeds.

Table 6: SAC+ML vs. TD3+BC

| Dataset | Environment | TD3+BC | SAC+ML |
|---|---|---|---|
| Random | HalfCheetah | $11.0_{\pm 1.1}$ | $17.3_{\pm 2.7}$ |
| | Hopper | $8.5_{\pm 0.6}$ | $7.9_{\pm 0.3}$ |
| | Walker2d | $1.6_{\pm 1.7}$ | $3.4_{\pm 2.1}$ |
| Medium | HalfCheetah | $48.3_{\pm 0.3}$ | $46.3_{\pm 0.2}$ |
| | Hopper | $59.3_{\pm 4.2}$ | $54.3_{\pm 3.4}$ |
| | Walker2d | $83.7_{\pm 2.1}$ | $81.2_{\pm 1.6}$ |
| Medium-Replay | HalfCheetah | $44.6_{\pm 0.5}$ | $42.5_{\pm 1.7}$ |
| | Hopper | $60.9_{\pm 18.8}$ | $48.8_{\pm 20.4}$ |
| | Walker2d | $81.8_{\pm 5.5}$ | $62.2_{\pm 4.9}$ |
| Medium-Expert | HalfCheetah | $90.7_{\pm 4.3}$ | $93.5_{\pm 4.0}$ |
| | Hopper | $98.0_{\pm 9.4}$ | $93.1_{\pm 7.8}$ |
| | Walker2d | $110.1_{\pm 0.5}$ | $108.1_{\pm 1.6}$ |
| Expert | HalfCheetah | $96.7_{\pm 1.1}$ | $93.6_{\pm 0.8}$ |
| | Hopper | $107.8_{\pm 7}$ | $110.9_{\pm 1.6}$ |
| | Walker2d | $110.2_{\pm 0.3}$ | $108.2_{\pm 0.3}$ |
| | Total | 1013.2 | 971.3 |

## D  ILLUSTRATION DISTRIBUTION SHIFT

Figure 5 shows the histogram of the $\ell_1$ norm of state vectors from the halfcheetah-medium-v2 and halfcheetah-expert-v2 datasets. Clearly, there is a distribution shift. So we can obtain in-distribution sample and out-of-distribution samples (with respect to the medium dataset) by (1) sample a transition from medium dataset and (2) sample from 98% to 100% percentile of the expert dataset (in terms of the $\ell_1$ norm of the state vector), so that it is out-of-distribution for a medium agent.

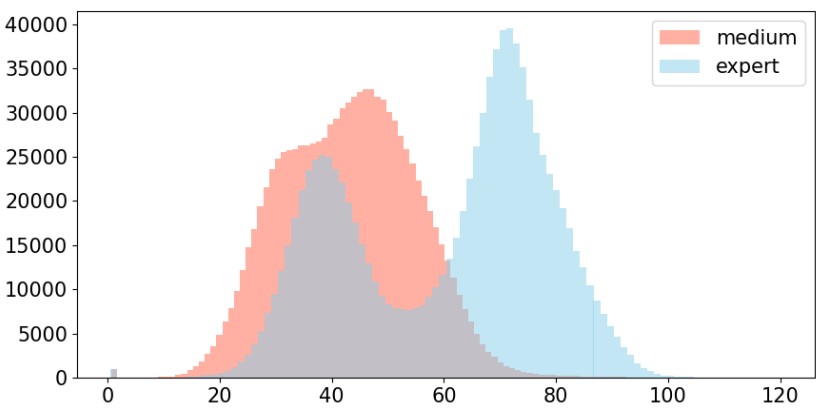

Figure 5: Histogram of $\ell_1$ norm of state vectors in halfcheetah medium and expert datasets.

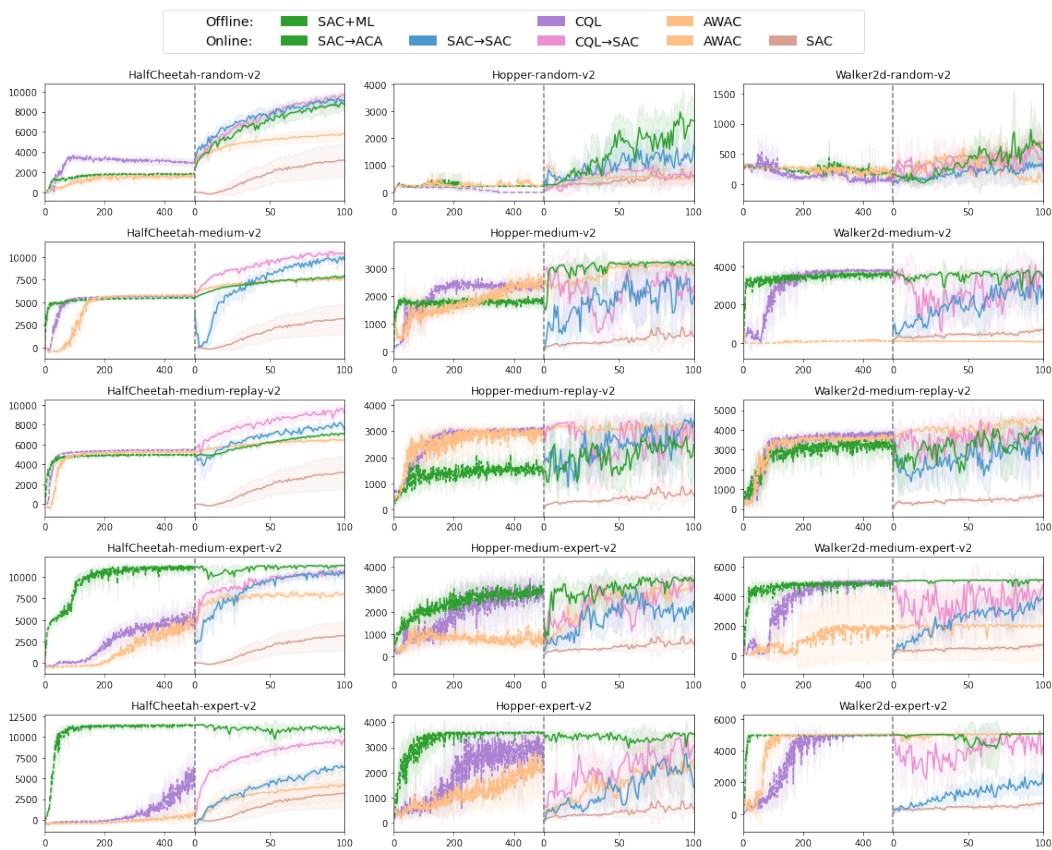

Figure 6: When offline data are not accessible, vs. other baselines

# E    ABLATION STUDY OF ACTOR-CRITIC ALIGNMENT

Here we write out the detailed formula of the critic objective in the two ablation studies in Section 6.5.

$$\mathcal{L}_Q^{\text{ablation1}}(\phi_i, \mathbf{d}) = \mathbb{E}_{(s,a,r,s',d)\sim\mathbf{d}}\left[\left(R_\phi(s,a) - y(r,s',d)\right)^2\right] \tag{32}$$

$$y(r,s',d) = r + \gamma(1-d)\mathbb{E}_{a'\sim\pi_\theta(\cdot|s')}\left(R_\phi(s',a') - \alpha\log\pi_\theta(a'|s')\right) \tag{33}$$

$$\mathcal{L}_Q^{\text{ablation2}}(\phi_i, \mathbf{d}) = \mathbb{E}_{(s,a,r,s',d)\sim\mathbf{d}}\left[\left(Q_\phi^{k,\beta}(s,a) - y(r,s',d)\right)^2\right] \tag{34}$$

$$y(r,s',d) = r + \gamma(1-d)\mathbb{E}_{a'\sim\pi_\theta(\cdot|s')}\left(Q_\phi^{k,\beta}(s',a') - \alpha\log\pi_\theta(a'|s')\right) \tag{35}$$

# F    MORE RESULTS ON ONLINE TRAINING WITHOUT OFFLINE DATA

The distributional shift issue would clearly be severer when offline data are not accessible during the online phase. To be more conservative, we therefore set $\beta_{w/o} = 1.5\beta_{w/}$ for experiments without offline data, excluding random and medium levels as both used no offline data for our main results already. ($\beta_{w/}$ denotes the hyper-parameter we used for our main results, see Table 8 for details.) All other hyper-parameters remained unchanged.

# G    SENSITIVITY ON $\beta$

Figure 7 shows that the performance of SAC→ACA is not very sensitive to the choice of $\beta$.

Figure 7: Results for different $\beta$

Besides, one could choose $\beta$ without any online iteration, rather than tuning by grid search, which is impractical in offline RL or O2O RL. One could compare the clipping threshold $|C_\beta(s)|$ to the magnitude of $|Z_\psi(s)|$, where $C_\beta(s)$ is as defined in Eq. (22).

$\beta$ is reasonable if these values are comparable, or $C_\beta(s)$ is slightly larger than than $|Z_\psi(s)|$. So the clipped $\log \pi_{\theta_0}$ term creates a barrier for the online critic to overcome, which in turn makes the transfer safe. This allows one to avoid running online evaluations to tune $\beta$, as the evaluation of $|Z_\psi(s)|$ and $|C_\beta(s)|$ can be done completely offline. We show the comparison between $|Z_\psi(s)|$ and $|C_\beta(s)|$ in Figure 8.

# H MAGNITUDE COMPARISON BETWEEN $|Z_\psi|$ AND $|C_\beta|$

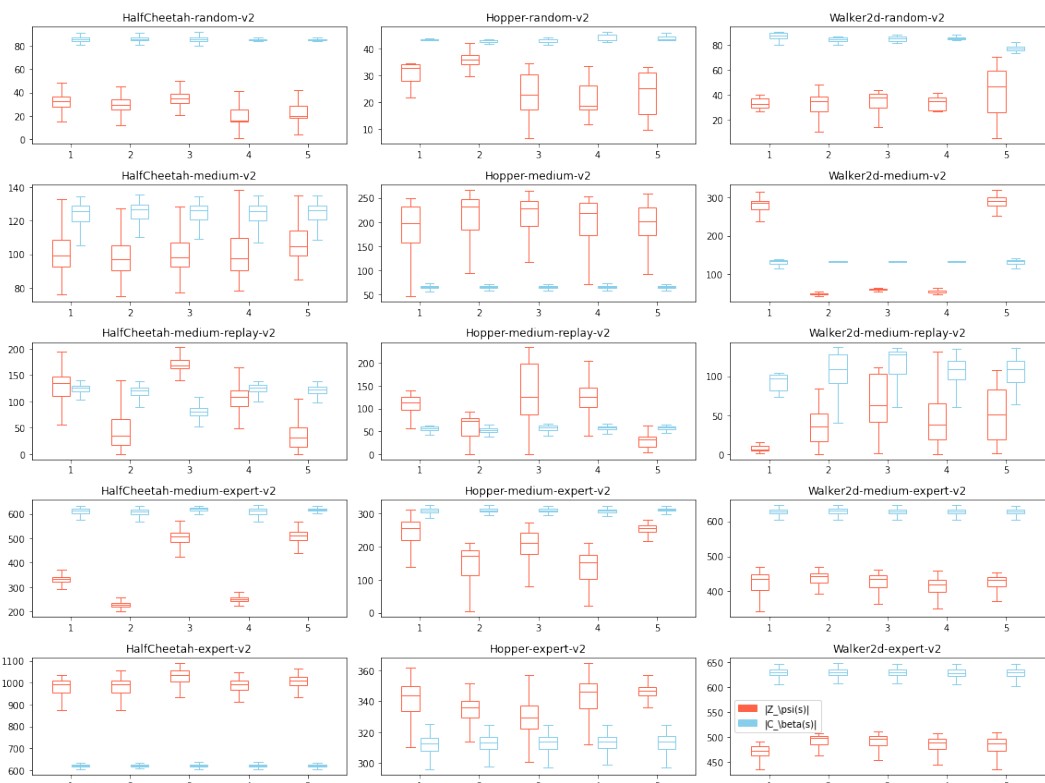

Figure 8: x-axis stands for different random seeds (corresponding to different pretrained SAC+ML models). For each random seed, we randomly sampled an episode $\{(s_i, a_i, s_{i+1}, r_i) : i = 1, 2, \ldots, T\}$ from corresponding offline dataset. We then computed $\{|C_\beta(s_i)|\}$ and $\{|Z_\psi(s_i)|\}$ to make box plots, where outliers were omitted for better visualization. We used the same $\beta$s as in Table 8 to make this plot.

Figure 8 shows that $|C_\beta(s)|$ and $|Z_\psi(s)|$ have comparable values. The empirical performance already outperforms all baselines even though we did not extensively match their magnitude for every task. It in turn implies that tuning $\beta$ requires minimal effort, in additional to the advantage that $\beta$ can be chosen completely via offline comparison.

# I   HYPER-PARAMETERS

Table 7: Specific hyper-parameters for different baselines. Please refer to the original paper for the meaning of hyper-parameter names.

| Algo. | Hyper-param name | Value |
|---|---|---|
| SAC+ML | $\omega$ | 30 |
| AWAC | $\lambda$ | 1.0 |
| CQL | conservative weight | 10 |
| | # of actions sampled | 10 |
| BR | offline buffer size | 2.5e6 |
| | online buffer size | 2.5e5 |
| | density ratio estimation network arch. | $[|\mathcal{S}| + |\mathcal{A}|, 256, 256, 1]$ |
| | density ratio estimation network temp | 5 |
| | $\rho$ | 0.75 |
| ACA | $\pi$ grad norm clip | 0.25 |

Table 8: Hyper-params used for our main results reported in section 6.1. $x \xrightarrow{y} z$ represents that $k$ decays from $x$ to $z$ using $y$ episodes.

| | hyper-param | HalfCheetah | Hopper | Walker2d |
|---|---|---|---|---|
| Random | N (# of init trajs) | | 0 | |
| | $\beta$ ($\beta$-clipping) | | 7 | |
| | k (interpolation) | | $1 \xrightarrow{10} 0$ | |
| Medium | N | | 0 | |
| | $\beta$ | | 7 | |
| | k | | $1 \xrightarrow{20} 0.5$ | |
| Medium-replay | N | | 50 | |
| | $\beta$ | | 7 | |
| | k | | $1 \xrightarrow{20} 0.5$ | |
| Medium-expert | N | | 50 | |
| | $\beta$ | | 15 | |
| | k | | $1 \xrightarrow{N/A} 1$ | |
| Expert | N | | 50 | |
| | $\beta$ | | 15 | |
| | k | | $1 \xrightarrow{N/A} 1$ | |

Table 9: General hyper-parameters. ACA and BR stand for SAC→ACA and CQL→BR, respectively, for the sake of space.

| | CQL | SAC+ML | ACA | SAC→SAC | BR | CQL→SAC | SAC (scratch) | AWAC (off) | AWAC (on) |
|---|---|---|---|---|---|---|---|---|---|
| Phase | offline | | online | | | | | offline | online |
| Based on SAC? | Yes | | | | | | | No | |
| *General hyper-params* | | | | | | | | | |
| $\pi$ Arch. | $[|\mathcal{S}| + |\mathcal{A}|, 256, 256, 1]$ | | | | | | | | |
| $Q$ Arch. | $[|\mathcal{S}| + |\mathcal{A}|, 256, 256, 1]$ | | | | | | | | |
| $Z$ Arch. | $[|\mathcal{S}|, 256, 256, 1]$ | | N/A for online | | | | | | |
| # $Q$ nets | 2 | | | | | | | | |
| # $Z$ nets | 2 | | N/A | | | | | | |
| $\tau$ (Polyak avg.) | 0.005 | | | | | | | | |
| Activation | ReLU | | | | | | | | |
| Optimizer | Adam for all | | | | | | | | |
| Adam params | betas = (0.9, 0.999), eps = 1e-8, weight decay = 0 | | | | | | | | |
| $\pi$ lr | 1e-4 | 3e-4 | 3e-4 | | | | | | |
| $Q$ lr | 3e-4 | 3e-4 | 3e-4 | | | | | | |
| $\alpha$ lr | 1e-4 | 3e-4 | 3e-4 | | | | | N/A | |
| $Z$ lr | 3e-4 | 3e-4 | N/A for online | | | | | | |
| # episodes | 500 | | 100 | | | | | | |
| # it/ep | 1000 | | | | | | | | |
| # batch/it | 1 | | | | | | | | |
| Batch size | 256 | | | | | | | | |
| *Hyper-params for the base SAC impl.* | | | | | | | | | |
| Entropy target $\mathcal{H}$ | $|\mathcal{A}|$ | | | | | | | N/A | |
| Uni-model Gaussian | Yes | | | | | | | N/A | |
| Squashed Gaussian | Yes | | | | | | | N/A | |

# J  IMPLICIT Q-LEARNING

We run IQL with the same experimental setting as Figure 2

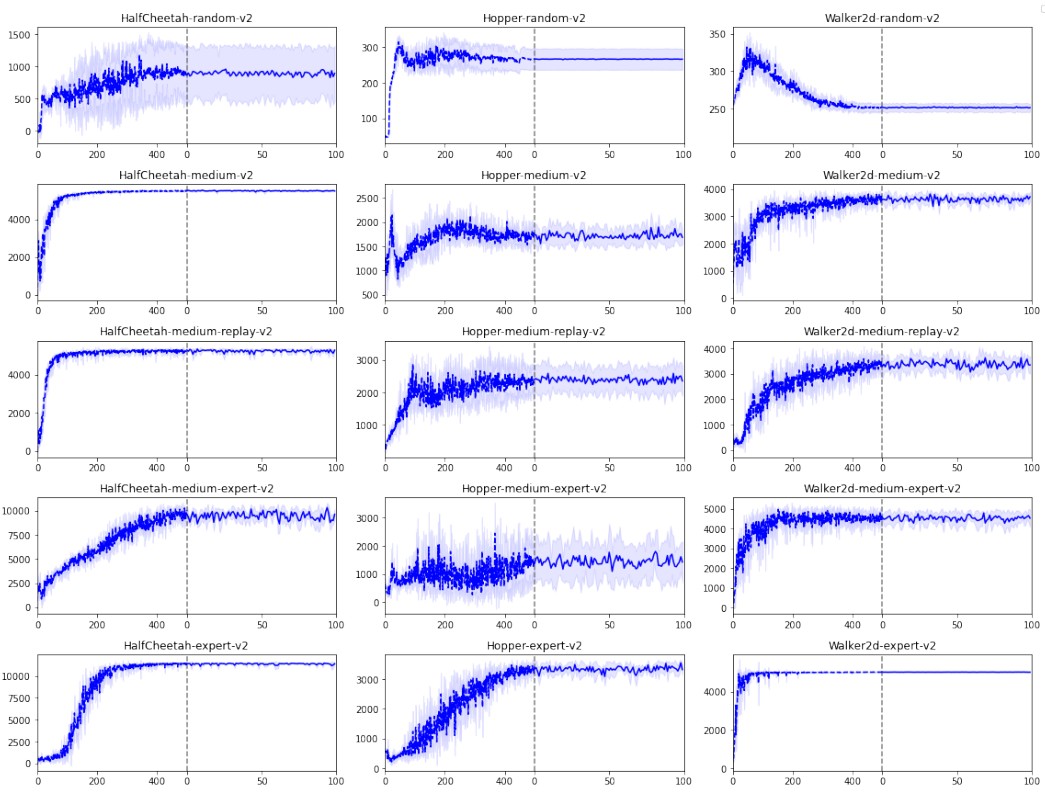

Figure 9: IQL

We observed that IQL is struggling with online improvement. This is in fact also observed by ODT (Zheng et al., 2022). See Table 10.

Table 10: IQL

| Dataset | Env | Reported by ODT | | | Our reproduction | | | | | |
|---|---|---|---|---|---|---|---|---|---|---|
| | | IQL(offline) | IQL(200k) | $\delta_{IQL}$ | IQL(offline) | IQL(100k) | $\delta_{IQL}$ | SAC+ML | ACA(100k) | $\delta_{ACA}$ |
| Random | HC | - | - | - | 9.37 | 9.45 | 0.07 | 17.29 | 72.60 | 55.31 |
| | H | - | - | - | 8.81 | 8.81 | 0.00 | 7.93 | 81.85 | 73.92 |
| | W | - | - | - | 5.46 | 5.44 | -0.02 | 3.36 | 12.42 | 9.06 |
| Medium | HC | 47.37 | 47.41 | 0.04 | 46.61 | 46.76 | 0.15 | 46.33 | 66.58 | 20.25 |
| | H | 63.81 | 66.79 | 2.98 | 55.62 | 52.48 | -3.14 | 54.31 | 96.54 | 42.24 |
| | W | 79.89 | 80.33 | 0.44 | 78.08 | 80.91 | 2.83 | 81.16 | 74.66 | -6.50 |
| Med.-Replay | HC | 44.10 | 44.14 | 0.04 | 44.40 | 44.85 | 0.45 | 42.53 | 59.03 | 16.50 |
| | H | 92.13 | 96.23 | 4.10 | 73.07 | 73.08 | 0.01 | 48.82 | 85.54 | 36.72 |
| | W | 73.67 | 70.55 | -3.12 | 74.64 | 73.03 | -1.61 | 62.19 | 85.17 | 22.98 |
| Med.-Expert | HC | - | - | - | 83.18 | 79.80 | -3.38 | 93.53 | 93.74 | 0.21 |
| | H | - | - | - | 44.61 | 44.42 | -0.19 | 93.08 | 98.02 | 4.94 |
| | W | - | - | - | 95.52 | 98.59 | 3.06 | 108.12 | 110.54 | 2.42 |
| Expert | HC | - | - | - | 94.07 | 94.10 | 0.03 | 93.59 | 93.14 | -0.46 |
| | H | - | - | - | 105.64 | 102.52 | -3.12 | 110.88 | 110.21 | -0.67 |
| | W | - | - | - | 108.98 | 109.10 | 0.13 | 108.21 | 109.59 | 1.38 |
| Total (med & med-replay) | | 400.97 | 405.45 | 4.48 | 372.42 | 371.11 | -1.30 | 335.35 | 467.53 | 132.19 |
| Total (all) | | - | - | - | 928.06 | 923.35 | -4.71 | 971.34 | 1249.65 | 278.31 |

## K  DEMONSTRATION OF MOTIVATION

How we design the experiment?

- We load pre-trained models from medium/expert level

- **Disable** critic update, target update, data collecting, etc. (In other words, only keep actor and entropy update.)

- The actor/entropy update are the same SAC actor/entropy update.

- Run update on random dataset to simulate OOD data

- We run total 10k steps and evaluate its performance every 100 steps.

to test how different offline/aligned $Q$-functions affects the actor update.

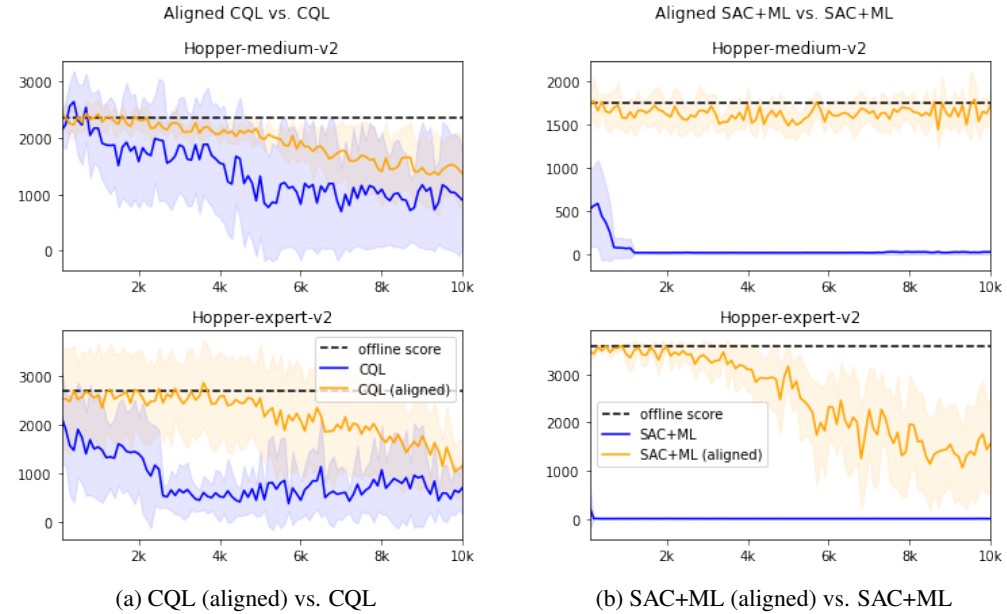

(a) CQL (aligned) vs. CQL       (b) SAC+ML (aligned) vs. SAC+ML

Figure 10: Our alignment (applied on both CQL and SAC+ML) could attain the offline performance better than the original performance.

**Takeaway 1**: our reparameterization is able to attain its offline performance better than other baselines.

**Takeaway 2**: our reparameterization applies to different offline critics, as we also mentioned in Section 6.3.

As SAC+ML collapse really quick, we opt it out for further comparison. We provide more comparison of CQL vs. aligned CQL.

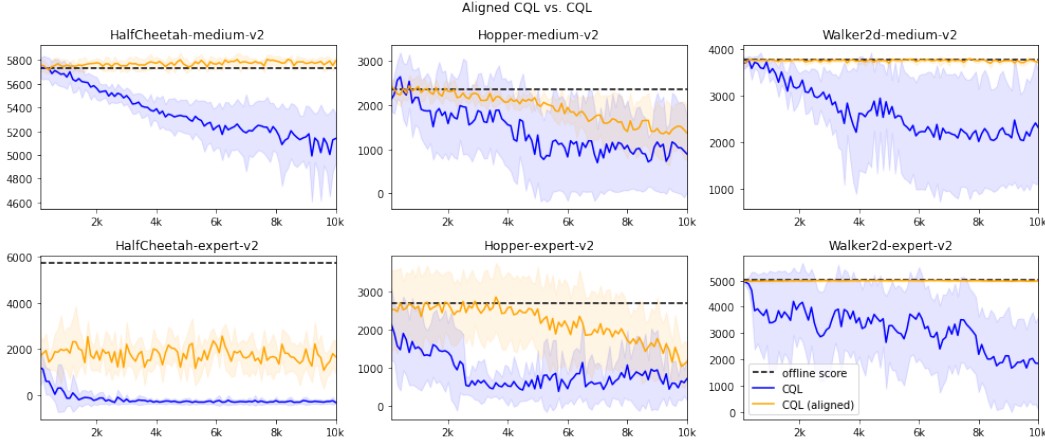

Figure 11: Aligned CQL vs. CQL.

Table 11: Expert level policies are more fragile.

| Dataset | Env | CQL relative change (%) | | | | | CQL (aligned) relative change (%) | | | | |
|---|---|---|---|---|---|---|---|---|---|---|---|
| | | @1k | @2k | @3k | @4k | @5k | @1k | @2k | @3k | @4k | @5k |
| Medium | HalfCheetah | -0.54 | -3.53 | -4.55 | -6.71 | -7.58 | 0.30 | 0.71 | 0.75 | 0.98 | 0.17 |
| | Hopper | -19.62 | -22.99 | -22.27 | -35.27 | -64.43 | -3.56 | -2.68 | -11.11 | -9.52 | -16.77 |
| | Walker2d | -5.22 | -11.74 | -25.76 | -30.89 | -26.73 | -0.12 | -0.09 | -0.40 | -0.70 | 0.01 |
| Avg. (medium) | | -8.46 | -12.76 | -17.53 | -24.29 | -32.91 | -1.13 | -0.69 | -3.59 | -3.08 | -5.53 |
| Expert | HalfCheetah | -102.50 | -102.32 | -105.22 | -105.48 | -104.96 | -68.42 | -73.67 | -68.21 | -69.02 | -62.44 |
| | Hopper | -44.96 | -51.07 | -77.36 | -80.78 | -82.59 | -12.83 | -5.23 | -9.44 | -3.31 | -19.33 |
| | Walker2d | -32.15 | -16.76 | -29.06 | -36.70 | -34.42 | -0.67 | -0.54 | -0.22 | -0.26 | -0.42 |
| Avg. (expert) | | -59.87 | -56.72 | -70.55 | -74.32 | -73.99 | -27.31 | -26.48 | -25.96 | -24.20 | -27.39 |
| Avg. (expert excluding HC) | | -38.55 | -33.92 | -53.21 | -58.74 | -58.51 | -6.75 | -2.89 | -4.83 | -1.79 | -9.87 |

**Additional observation**: near optimal policies are more fragile (even if we exclude HalfCheetah-expert), which highlights our advantage in expert-level datasets.

## L   AWR OBJECTIVES

Akin to experiments we conducted in Section K, we run actor update only experiments with AWAC (also categorized into AWR) actor objective instead of SAC actor objective. Critics are trained offline by AWAC.

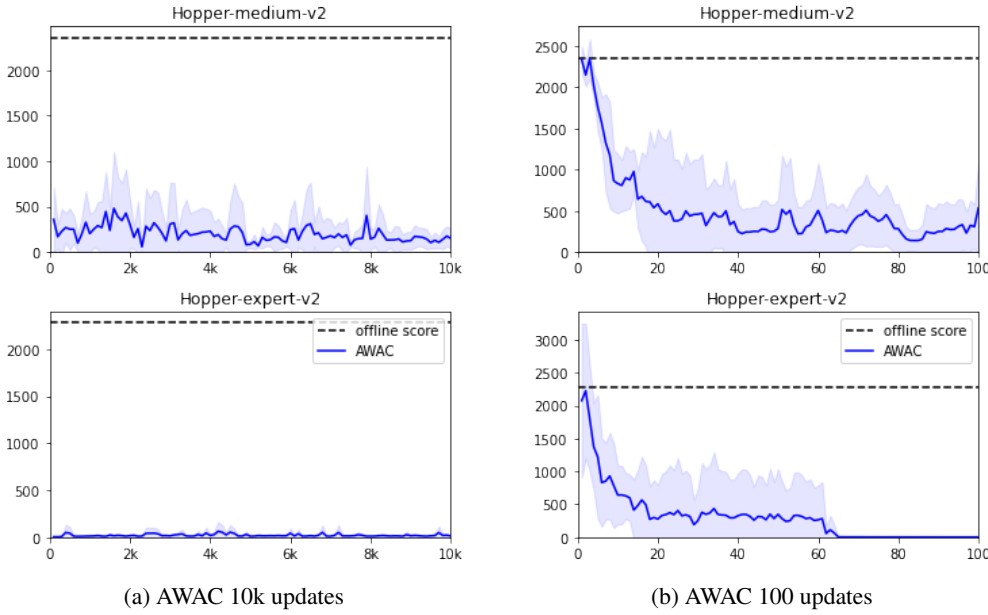

(a) AWAC 10k updates                     (b) AWAC 100 updates

Figure 12: (a) shows that AWAC also collapse to nearly zero performance. (b) shows that around 20 actor updates are already enough to destroy the performance.

## M    AVERAGED VERSION OF FIGURE 1

Figures 14 and 15 provide averaged versions of Figure 1 for in-distribution and out-of-distribution samples, respectively. The approach used to generate these two figures is illustrated in Figure 13.

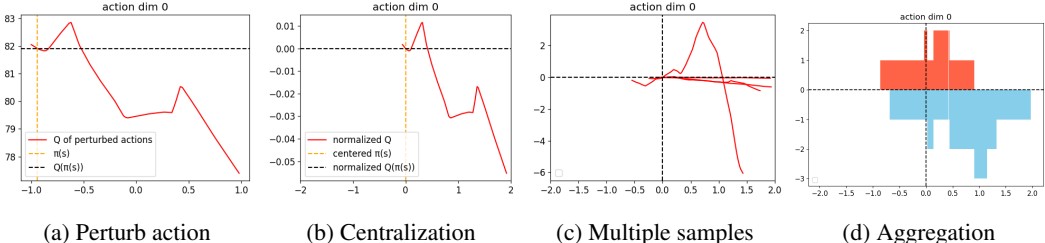

(a) Perturb action          (b) Centralization          (c) Multiple samples          (d) Aggregation

Figure 13: Demonstration of how Figure 14 and Figure 15 are created. (a) Given a sample $(s, a)$, we perturb $a$ along a dimension to plot $Q(s, \tilde{a})$ and compare $Q(s, \tilde{a})$ to $Q(s, \pi(s))$; (b) We plot $Q(s, \tilde{a})$ with its deviation from $Q(s, \pi(s))$, so that $Q(s, \pi(s))$ is centered at $y = 0$ and $\pi(s)$ is centered at $x = 0$; (c) Such a centralization allows us to place multiple samples (different $s$) in the same plot, where points above the $x$-axis correspond to "over-estimated" perturbations; (d) We aggregate multiple samples by counting how many points are above 0. This way, the height of the red part in the bar plot quantifies the fraction of points that are "over-estimated". For an "over-estimated" point $(s, a)$, its $x$-coordinate stands for the distance between $a$ and the policy favored action $\pi(s)$.

**Note**: By "over-estimation", we mean for some $a \neq \pi(s)$ such that $Q(s, a) > Q(s, \pi(s))$, which in a certain degree means that the critic $Q$ is "disagreeing" with the policy $\pi$.

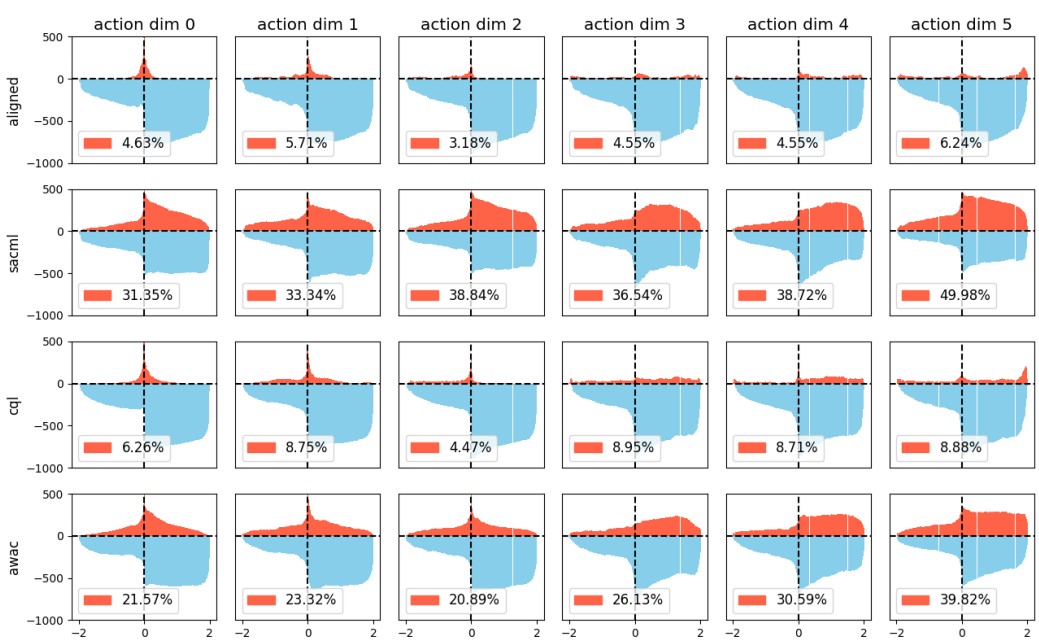

Figure 14: Quantifying fraction of over-estimated perturbations for in-distribution samples.

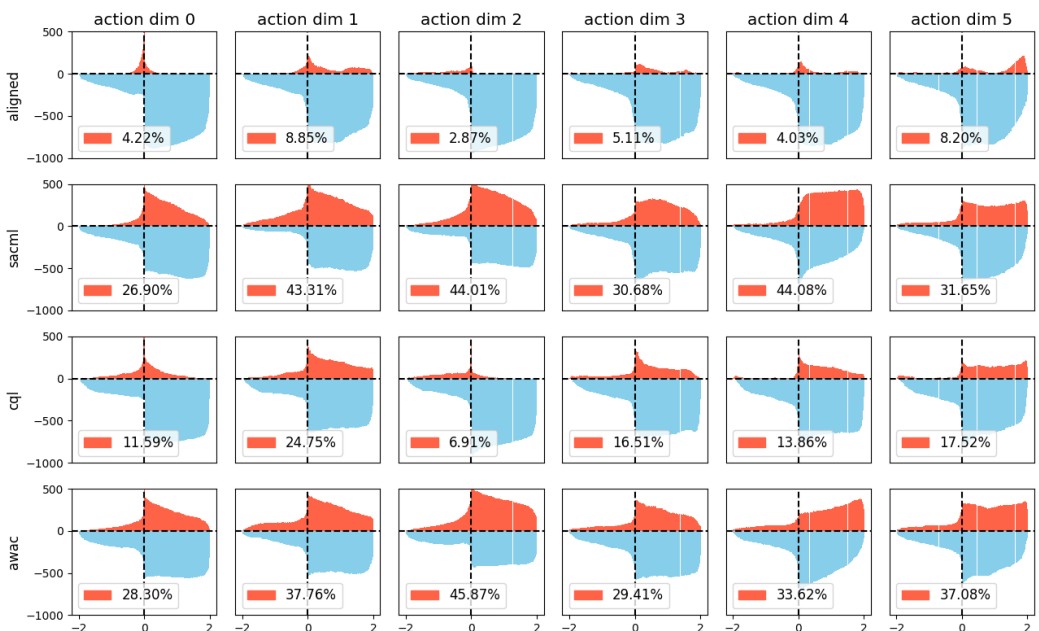

Figure 15: Quantifying fraction of over-estimated perturbations for out-of-distribution samples.

**Details**: All agents are trained on halfcheetah-medium-v2 dataset. By in-distribution samples, we refer to states drawn from the halfcheetah-medium-v2 dataset. By out-of-distribution samples, we use samples from the halfcheetah-expert-v2 dataset. We randomly drew 200 samples per seed, which results in a total of 1000 samples to make each plot.

## N  CONVERGENCE OF SOME TASKS

We run another 1M steps on top of the original 500k steps, for HalfCheetah-Medium-Expert, HalfCheetah-Expert, Hopper-Expert, with experimental setting and hyperparams that are identical to Figure 2.

However we found that HalfCheetah-Medium-Expert still have high variance. CQL suffer from relatively high variance is also observed in TD3 (Fujimoto & Gu, 2021).

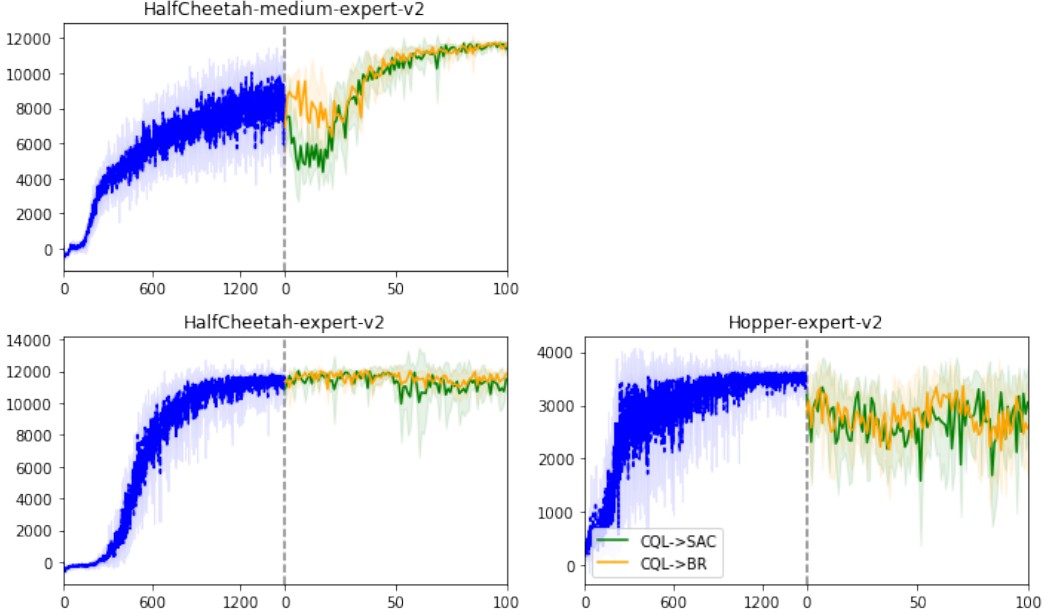

Figure 16: Run CQL until convergence.

We still observe unstable performance in Hopper-expert. Due to time limit, we are not able to run HalfCheetah-medium-expert further longer, but it should not affect the results in Table 12 as the online performance of HalfCheetah-medium-expert converged (roughly) to max score.

Table 12: New results with additional 1M CQL offline steps

| Task | CQL→SAC (prev.) | CQL→SAC (new) | CQL→BR (prev.) | CQL→BR (new) | Ours |
|---|---|---|---|---|---|
| HC-ME | 87.85(46.77) | 93.94(28.75) | 91.80(50.73) | 96.35(31.17) | 93.74(0.21) |
| HC-E | 55.39(6.86) | 94.88(-0.98) | 79.69(31.16) | 97.67(1.82) | 93.14(-0.46) |
| H-E | 67.88(-15.48) | 94.56(-13.95) | 68.55(-14.81) | 79.32(-29.19) | 110.21(-0.67) |
| Total (three above) | 211.12(38.15) | 283.37(13.83) | 240.04(67.08) | 273.34(3.79) | 297.09(-0.92) |
| Total (all) | 1016.08(79.35) | 1088.33(55.03) | 1190.96(254.23) | 1224.26(190.94) | 1249.65(278.31) |

As originally we claim we match BR's performance, and we highlighted both BR and ours as the strongest ones in Table 2. We do not see a nessceray change of this claim. The new results is in fact favoring us in terms of performance improvement. Our ultimate advantage over BR is that ours could attain similar performance while offline data is not accessible while BR is not even applicable.

## O    INITIALIZATION FROM BEST CHKPTS

We only saved checkpoints **every 100 episodes**. Therefore, we pick the best checkpoints among them. Sub-figures that are not visible means the last checkpoint is the best checkpoint we saved. (We determine the "best" checkpoint by the averaged performance over all seeds.)

We do **not** observe significant differences between initialization from best checkpoints and last ones.

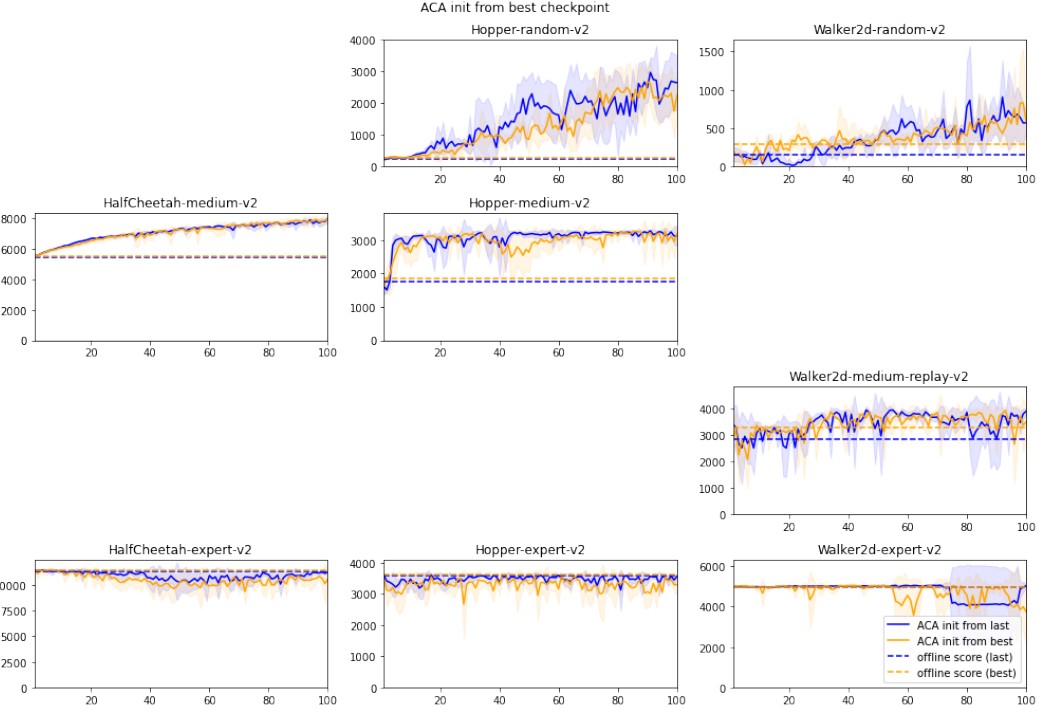

Figure 17: ACA initialized from best checkpoint, if last is not the best.

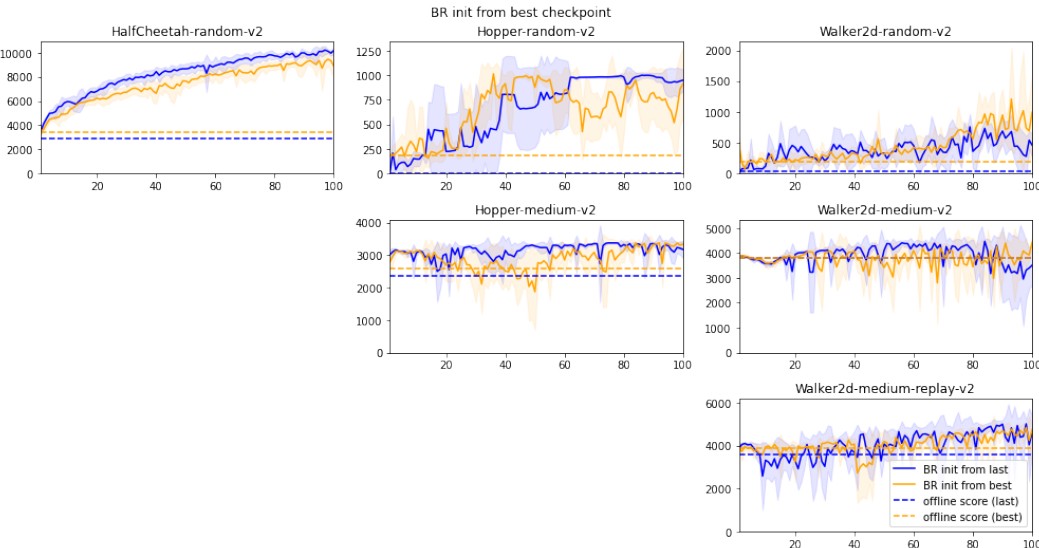

Figure 18: BR initialized from best checkpoint, if last is not the best.

## P  DOES OVER-FITTING AFFECT TRANSFER?

In addition to Section O, we initialize from checkpoints at 100k, 200k, 300k, 400k and 500k respectively for walker2d-medium-v2, to see whether initialization from different checkpoints make a difference.

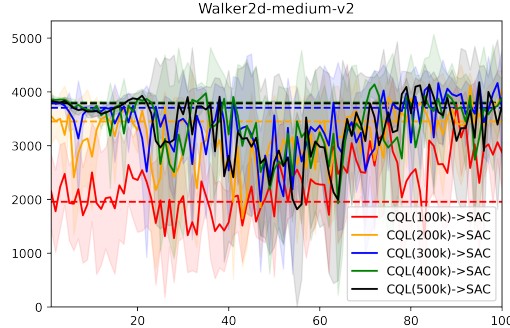

Figure 19: CQL→SAC initialized from different checkpoints. Dashed lines are corresponding offline performance.

We do not observe conclusive evidence showing that over-fitting might lead unstable transfer.

## Q  ASYMPTOTIC PERFORMANCE

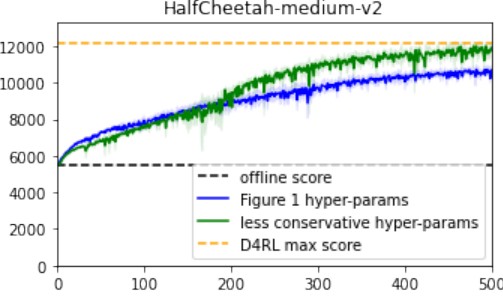

Figure 20: Asymptotic performance

The blue curve uses the same hyper-parameter setting to Figure 2, where $k$ decays from 1.0 to 0.5 in 20k steps. Keeping $k = 0.5$ seemed to slow down the achievement of expert performance. However, one could always safely to decay $k$ to 0 after the transfer is stable. The green curve linearly anneals $k$ to 0 using 200k steps and is able to reach D4RL expert score much faster.

# R  RANDOM INITIALIZATION

We initialize $R(s, a)$ randomly

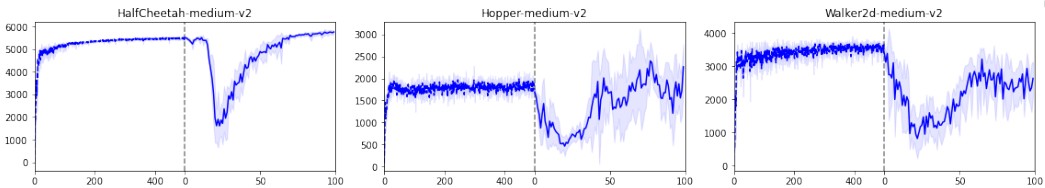

Figure 21: Randomly initialized $R(s, a)$

It is obvious that random initialization has poor performance as it violates our motivation.

