# OpenReview forum: "Actor-Critic Alignment for Offline-to-Online Reinforcement Learning"
_ICLR.cc/2023/Conference — Submitted to ICLR 2023_

### Official Review · Reviewer_Xgqq · 2022-10-24

**Confidence:** 4
**Correctness:** 2
**Technical Novelty And Significance:** 2
**Empirical Novelty And Significance:** 3
**Recommendation:** 5

**Clarity, Quality, Novelty And Reproducibility:**

I think the paper is not clear enough to show the necessity of algorithm design choice. The originality of the method is also a bit limited as it highly builds upon TD3+BC.

**Strength And Weaknesses:**

Strength:

1. The paper tackles offline-to-online finetuning, which is an important problem to study in RL.

2. The empirical results suggest that proposed method is able prevent dip at the start of the fine-tuning, which is promising.

Weaknesses:

1. I'm confused by the fact that the authors need to do explicit alignment at the end of offline training as the added BC regularization should be able to handle distributional shift of Q-values on out-of-distribution actions. It is unclearly why the authors need to do this additional step. It appears to me that this step is rather ad-hoc.

2. The authors also didn't compare to more recent offline-to-online methods such as IQL. It is important to add such a comparison to show the benefits of the method. It also appears that the improvement over other approaches is rather marginal, which is not conclusive enough to show the benefits of the method.

**Summary Of The Paper:**

This paper presents a new offline-to-online RL method that proposes to use SAC as the backbone RL algorithm and directly aligns the policy with the critic function at the end of the offline training phase and use the newly aligned critic as an initialization of the critic function during online fine-tuning. The authors show that the proposed approach can achieve reasonable performance on D4RL-v2 tasks.

**Summary Of The Review:**

Based on the comments above, I would vote for a weak reject.

---

> ### Author Response · Authors · 2022-11-18
> **Subject: Response to Reviewer Xgqq**
>
> We thank the reviewer for the constructive review. We provide a detailed response to your concerns below.
>
> **Q1:** Clarification on motivation
>
> **A:** We would like to stress that our work is not ad-hoc, but originates from a principled motivation.
>
> - Why Eqn (5)?
>
>     The motivation behind the definition of $Q_\mu$ in Eqn (5) is as follows. Eqn (4) is the closed-form minimizer of entropy-regularized actor loss, i.e. Eqn (1).  So how should $Q_\mu$ look like such that when it is plugged into the right-hand side of Eqn (4), it will exactly match the $\pi_\theta$ in the left-hand side?  It is not hard to see that such a $Q_\mu$ must be in the form of Eqn (5) for some $Z(s)$.
>     As a result, updating $\pi$ based on the value of $Q_\mu$ computed from Eqn (5) produces the same policy $\pi_\theta$ in theory, hence achieving our goal of keeping the policy transfer safe in the initial online stage.
>
>
> - Why initialize $R(s, a)$ with $Z(s)$?
>
>     Initializing $R(s, a)$ with $Z(s)$ ensures that the online $Q$-function is initially consistent with the offline policy, while achieving low Bellman residual.  This proffers the desired safe transfer.  However, we also refrain from constraining the $Q$-function to the closed-form manifold, allowing more flexibility for the online improvement of both the policy and the $Q$-function. To summarize, initializing $R(s, a)$ with $Z(s)$ is our practical solution to benefiting from safe transfer but not being overly restrictive.
>
>
> This behavior cloning regularization is naturally induced from this closed-form minimizer, rather than an ad-hoc heuristic design. Hence, we do not regard our algorithm as built upon TD3+BC, although we use SAC+ML for offline training. The alignment step is our contribution rather than SAC+ML. We also showed in Section 6.3 that ours can be warm-started from CQL instead of SAC+ML.
>
> To demonstrate our motivation, we conducted additional experiments in Section K, where we ran only actor updates with offline-trained and aligned critics on out-of-distribution data, disabling other aspects such as critic update and data collecting. Our method was able to best attain the offline policies' performance, as expected from our theoretical motivation. Besides, we aligned both CQL critic and SAC+ML critic in Section K, which also show that our proposed alignment is quite orthogonal to TD3+BC.
>
>
> **Q2:** IQL performance
>
> **A:** We did not include IQL because Zheng et al. (2022) have already shown that ODT outperform IQL, and we included ODT for comparison. In fact, Zheng et al. (2022) show that IQL provides very restricted policy improvement in MuJoCo locomotion tasks (+1.1\%). We provide our reproduction results of IQL in Section J, with similar observation (-0.5\%).
>
> We used the word "match" in our claim and also highlighted both BR and our method in Table 2. In addition, we further discussed our advantages in Section 6.4 and Section 6.3. Section 6.4 shows that ACA could achieve similar performance without accessing offline data during online stage while BR is not applicable. Section 6.3 shows that BR critically relies on CQL while ours can be initialized from different choices and achieve similar performance.

---

### Official Review · Reviewer_XAHs · 2022-10-29

**Confidence:** 4
**Correctness:** 2
**Technical Novelty And Significance:** 4
**Empirical Novelty And Significance:** 2
**Recommendation:** 5

**Clarity, Quality, Novelty And Reproducibility:**

### Clarity, Quality
Most of the part is clearly written and understandable, but there are a few unclear parts (described in the prior section).

### Novelty
The proposed method is novel.

### Reproducibility
Implementation code and a complete set of hyper-parameters are provided for reproducing.

**Strength And Weaknesses:**

### Strength
* The proposed method can achieve comparable online convergence with SOTA baselines without its access to the offline data or importance sampling strategy.

### Weakness
1. Unclear parts in experiment results.
    - Some of the CQL offline training curves in Figure 2 seem to be not converged yet. (HC medium-expert, HC expert, and H expert, following the notation in table 2). It could have affected the initial stability and also the final score after a fixed amount of online training, which makes the quantitative comparison in Table 2 unreliable.
    - In Figure 1 result, SAC+ML is not even aligned with the in-distribution sample which is considered to be from the bias introduced by ML term. (or possibly from not converged Q which is also problematic.) Thus, this qualitative result is showing the misalignment not only by distributional shift but also by biased objective. Consequently, the given result is not enough to illustrate the existence of the misalignment problem of prior offline RL methods.

2. Ambiguous term - misalignment
    - Despite the "actor-critic misalignment" being the key problem that this paper aims to solve, it is neither formally defined nor kindly explained.
    - Moreover, as described in weakness-1, the qualitative result is not clearly showing whether the described problem really appears in other methods.

3. IQL reports better benchmark results than AWAC even for online fine-tuning, thus more proper baseline for AWR-based offline RL methods. I think this comparison is also important to show whether AWR policy objective is also affected by the actor-critic misalignment. But the current bad results of AWAC seem to be from offline RL performance rather than the actor-critic mismatch.

**Summary Of The Paper:**

The paper proposes a novel offline-to-online RL method.
The proposed method introduces an offline critic re-training phase to explicitly align the critic with the offline-trained policy.
In the critic re-training phase, a refined critic parameterization deducted from SAC policy objective prevents value overestimation.
Experimental evaluation shows its comparative fine-tuning sample efficiency compared to SOTA offline-to-online RL methods.

**Summary Of The Review:**

The proposed method and empirical support are quite convincing.
ACA can replace the SOTA offline-to-online RL method even without access to offline data.
However, as described in the weaknesses, some important points related to the motivation of the idea are not clearly shown.
Also, the quantitative comparison possibly includes unfair comparisons for several tasks.
Thus, I vote for rejecting this paper until those concerns are resolved.

---

> ### Author Response · Authors · 2022-11-18
> **Subject: Response to Reviewer XAHs**
>
> We thank the reviewer for the constructive review. We provide a detailed response to your concerns below.
>
> **Q1:** Some runs do not converge
>
> **A:** We ran halfcheetah-medium-expert, halfcheetah-expert and hopper-expert for an additional 1M offline steps. The new results could be found in Section N. The final performance of BR increased by around 30 and the performance $\delta$ of BR dropped by around 60. Indeed, such a result is slightly in favor our method in terms of finetuning improvement.
>
> But overall, we do not see the new experiments make any difference to the conclusion. We originally used the word "match" in our claim and highlighted both BR and ACA in Table 2.
>
> This does not affect our advantages that (a) as mentioned in Section 6.4, ACA attains similar performance even when offline data is not accessible during online stage, a scenario where BR is inapplicable; (b) ACA can be initialized from a broader choice of offline learners while BR's performance critically relies on CQL, as shown in Section 6.3.
>
>
>
> **Q2:** Does ``misalignment'' exist in other baselines?
>
> **A:** We agree it is important to give more comprehensive study on top of Figure 1. We therefore, gives a more rigorous version of Figure 1 in Section M, and designed additional experiments to verify our motivation in Section K.
>
>
> - In Section M, we measure how many $(s, a)$ pairs admit the behavior that $Q(s, a) > Q(s, \pi(s))$, which in certain way reflects that how the $Q$-function "disagreeing" with $\pi$. We term $Q(s, a) > Q(s, \pi(s))$ as "over-estimation". Details on how we measure the fractions and make the plots can be found in Figure 13.
>
>     We observed that all offline baselines, SAC+ML, CQL, AWAC presents larger fractions of "over-estimated" actions. Our aligned critic is able to achieve the best for both in-distribution samples and out-of-distribution samples.
>
>     CQL in fact comes as close second for in-distribution samples. This also confirms that CQL has better performance compared to other baselines in Table 2. However, for out-of-distribution samples, our re-parameterized $Q$-function admits a much smaller fraction of "over-estimation", compared to CQL. Informally put, our alignment could significantly address such "disagreement" between $Q$-function and $\pi$, and this issue is not only appeared in SAC+ML.
>
> - In addition, we designed additional experiments where we run only actor updates on out-of-distribution data with offline-trained/aligned $Q$-function respectively. Informally put, if a $Q$-function is consistent with a policy $\pi$, running the actor update should attain $\pi$'s performance. However, we observed that using SAC+ML, CQL critic to update the corresponding policy $\pi$ leads to performance degradation, even though all other updates are disabled. Our aligned $Q$-function did the best job to attain performance of $\pi$.
>
>     Results could be found in Section K. We observed that CQL drops significantly slower than SAC+ML, this conform that CQL has fewer over-estimations that observed in Section M.
>
> We believe these two additional observations could show that this issue exists in other baselines than SAC+ML.
>
> **Q3:** IQL performance
>
> **A:** We did not include IQL because Zheng et al. (2022)  have already shown that ODT outperform IQL, and we included ODT for comparison. In fact, Zheng et al. (2022) show that IQL provides very restricted policy improvement in MuJoCo locomotion tasks (+1.1\%). We provide our reproduction results of IQL in Section J, with similar observation (-0.5\%).
>
>
> **Q4:** AWR objective affected?
>
> **A:** As shown in M, AWAC's $Q$-function ``over-estimates'' a lot of actions. In addition, we ran a similar actor-only experiments with AWAC (also categorized into AWR) actor objectives in Section L. It can be observed that AWAC is also not able to attain the offline policy performance. In fact, it only took around 20 updates to destroy the policy performance with all other factors kept intact.

---

### Official Review · Reviewer_Px1x · 2022-10-31

**Confidence:** 5
**Clarity, Quality, Novelty And Reproducibility:** The paper is not clearly written and …
**Correctness:** 3
**Technical Novelty And Significance:** 2
**Empirical Novelty And Significance:** 2
**Recommendation:** 3

**Strength And Weaknesses:**

Strength:

- Utilizing a trained model in the offline-rl and fine-tuned during online learning is an interesting direction to consider.
- Some of the results are promising.


Weaknesses:

- Although authors claim their method "outperforms or matches the current SOTAs", I disagree with this statement. Take table 2 and Figure 2 as examples. In table 2, CQL + BR has pretty much the same performance as SAC + ACA ( proposed method). Although standard  deviation is not reported in these experiments, it is hard to say if these results are significant. In addition, if we look at Figure 2, green lines are only better in 5 to 6 cases out of 15 cases. Considering the strong performance of [1], why should someone select this method over [1]? What is the advantage of this method vs [1]? discarding Q-function by itself is not an advantage until it shows up in the results.

- This method has lots of moving parts and it would be potentially hard to make if this method applies to new environments.  Consider training settings for each step of this method.

- It is hard to follow the writing and understand the motivation for some choices. For instance, $Z$ is a function of only states but during the final step of this algorithm, it will be used to initialize R(s, a). Since R(s,a) is a function of both s and a, why does this initialization make sense? Also, it is not clear the motivation behind  $Z$. Can you explain?  It seems to me $Z$ behaves like a value function normalized by policy. But it is hard to tell. Also, can you explain how you derive equation 5 and motivation beyond that? Finally what happens if we ignore the first two steps of your method and just initialize R(s,a) randomly? How does performance change?

- In my view, the issue of distribution shift is overlooked in this paper and there are no good cases for that in this work. Examples of distribution shift can be change of state distribution during online training, changing reward, changing transition functions, etc.

- Why were only these environments selected? Are there reasons for not running experiments using Adroit, Maze environment form D4RL benchmarks?

Other comments:
- Compounding error is usually the problem in model-based RL. Can you expand and discuss how it is related to offline RL?

- Extrapolation error is mainly the source of the issues in offline RL. However, this paper implies that it is a distribution shift. Since not all extrapolations are caused by distribution shift and vice versa. Can you explain how these two are related?


[1] Seunghyun Lee, Younggyo Seo, Kimin Lee, Pieter Abbeel, and Jinwoo Shin. Offline-to-online reinforcement learning via balanced replay and pessimistic q-ensemble.

**Summary Of The Paper:**

This paper proposes a way to transfer (or align ) a trained method from offline learning to online learning. This method has three steps: (modified) offline learning, actor-critic alignment, and online training. In  modified  offline learning, they use a modified version of TD3-BC where they use soft-actor-critic instead  of TD3 and add behavior cloning term. During the alignment step, they fit the "baseline", and finally during online-training, they changed the Q function with their "baseline". To show the effectiveness of their method, they use a subset of D4RL benchmark (i.e HalfCheetah-*, Hopper-*, Walker2d-*) and they get interesting results.

**Summary Of The Review:**

Offline to online alignment/transfer in RL is an important and interesting area and this paper looks at the right problem. However, the current paper has many issues and I don't think it meets ICLR acceptance bar. In addition, the paper doesn't experiment in which distribution shift really happens, this is important as this paper motivates the issue of offline to online transfer. Finally, it requires significant revision as it is hard to follow.

---

> ### Author Response · Authors · 2022-11-18
> **Subject: Response to Reviewer Px1x (Part 1)**
>
> We thank the reviewer for the constructive review. We provide a detailed response to your concerns below.
>
> **Q1:** Our advantage
>
> **A:** We think our claims are accurate.  In particular, we used the word "match" (first bullet above Section 6.1) and highlighted both CQL$\to$BR and ours in Table 2. We also explicitly highlighted our advantages over BR in Section 6.3 and 6.4.
>
> Section 6.3 shows that BR's performance critically relies on the choice of offline training algorithm. And ours can be initialized from different choices without much performance difference. This is our first advantage over BR, i.e. when offline training is out of our control.
>
> Section 6.4 highlights an even more significant advantage -- our method could attain nearly the same performance without accessing offline data during the online stage. In this scenario, BR is not applicable. And this is in fact a very realistic scenario especially when privacy is an important concern.
>
> Incidentally, we do not see discarding $Q$-function as an advantage, it is simply a way to implement our theoretical motivation, which should be considered as another merit.
>
>
> **Q2:** Convenience of deployment
>
> **A:**
> Our reparameterization is arguably less expensive in computation as we do not compute, e.g., KL divergence constraints or on-policy-ness of samples. Moreover, we do not think it introduced more hyper-parameters than BR and is easier to implement.
>
>
> **Q3:** Clarification on motivation
> - (Why Eqn (5)?)
>
>     The motivation behind the definition of $Q_\mu$ in Eqn (5) is as follows. Eqn (4) is the closed-form minimizer of entropy-regularized actor loss, i.e. Eqn (1).  So how should $Q_\mu$ look like such that when it is plugged into the right-hand side of Eqn (4), it will exactly match the $\pi_\theta$ in the left-hand side?  It is not hard to see that such a $Q_\mu$ must be in the form of Eqn (5) for some $Z(s)$.
>     As a result, updating $\pi$ based on the value of $Q_\mu$ computed from Eqn (5) produces the same policy $\pi_\theta$ in theory, hence achieving our goal of keeping the policy transfer safe in the initial online stage.
>
>     We designed additional experiments to demonstrate this motivation.  In Section K, we ran offline trained actor-critic agents on out-of-distribution data with actor update only. Our re-parameterized $Q$-functions could attain the offline policies' performance, which verifies the effectiveness of our theoretical motivation, leading to a safe transfer.
>
> - (Why initialize $R(s, a)$ with $Z(s)$?)
>
>   Initializing $R(s, a)$ with $Z(s)$ ensures that the online $Q$-function is initially consistent with the offline policy, while achieving low Bellman residual.  This proffers the desired safe transfer.  However, we also refrain from constraining the $Q$-function to the closed-form manifold, allowing more flexibility for the online improvement of both the policy and the $Q$-function. To summarize, initializing $R(s, a)$ with $Z(s)$ is our practical solution to benefiting from safe transfer but not being overly restrictive.
>
> - (Is $Z(s)$ a value function?)
>   $Z(s)$ is the value function plus entropy, because
>   $V(s) \coloneqq \mathbb{E}_\pi Q(s, a) = \mathbb{E}_\pi[Z(s) + \alpha\log\pi(a|s) ] = Z(s) - \alpha\mathcal{H}(\pi(\cdot|s)).$
>
> - (Randomly initialize $R(s, a)$)
>
>   We conducted additional experiments with randomly initialized $R(s, a)$.  As shown in Section R, with no surprise, it performs poorly as it does not follow our motivation.
>
>
> **Q4:** Clarification on distribution shift.
>
> **A:** We agree that ideally, distribution shift in RL can cover change of state distribution during online training, changing reward, changing transition functions.  Analogous to transfer learning where covariate and label distributions can shift across domains, the MDPs themselves can change in offline-to-online RL.  Although this is an interesting direction, our paper follows a cohort of recent literature by focusing on the shift of the state/action distributions between offline dataset and state/action distribution encountered online. For example, this term is also used in Balanced Replay (Lee et al., 2022) to describe state/action distribution mismatch between offline and online, without any change of the environment/MDP; see their Figure 1(a) for example.
>
> As such, we think our experiments naturally reflect this problem because at the online stage,
> the learned policy deviates from the offline behavior policy, hence shifting the state/action distribution compared with the offline dataset. Besides, the additional experiments in Section K that only update actors make the distribution shift problem more explicit, where we train agents (which are learned from medium/expert dataset) on random dataset to explicitly simulate change of state-distribution.

---

> > ### Author Response · Authors · 2022-11-18
> > **Subject: Response to Reviewer Px1x (Part 2)**
> >
> > **Q5:** Additional environments
> >
> > **A:** The main reason we opted out AntMaze and Adroit is because there are only a few cases where both TD3+BC (hence SAC+ML) and CQL achieve decent offline score; if offline scores are low, safe transferring is meaningless. For AntMaze, the only level that both TD3+BC and CQL works is u-maze (for example, reported by [6] and IQL [1]). For Adroit, the only scenario where both TD3+BC and CQL works is Pen (for example, reported by [2]). Therefore, one could not carry out a complete benchmark anyway. So we opted out for these environments.
> >
> >
> > **Q6:** Clarification on compounding error
> >
> > **A:**
> >
> > Compounding error is not necessarily an unique term used in model-based RL although it is commonly seen in there. In the context of offline RL, [3] states "to reduce the effect of *compounding errors issue in offline RL*", and [4] states "Prior offline model-free RL algorithms, ... rely on estimating Q-values for optimizing the target policy. This procedure often yields unreasonably high Q-values due to the **compounding error** from bootstrapped estimation with out-of-distribution actions ..."
> >
> > Since our alignment explicitly tackles such an over-estimation problem raised by bootstrapping with out-of-distribution actions, we believe it effectively addresses the issue of compounding error.
> >
> >
> > **Q7:**  Clarification on extrapolation error
> >
> > **A:**  According to [5], extrapolation error refers to ``a phenomenon where unseen state-action pairs are erroneously estimated to have unrealistic values. Extrapolation error can be attributed to a mismatch in the distribution of data induced by the policy and the distribution of data contained in the batch.''
> >
> > Since we keep the MDP intact while the learned policy deviates from the behavior policy, the distribution of data induced by the online policy clearly mismatches that in the offline data,
> > which leads to the extrapolation error as pointed out by [5].
> > Conversely, extrapolation error incorrectly promotes certain actions, which in turn exacerbates the shift of state/action distribution.
> >
> > Our actor-critic alignment alleviates both distribution shift (as discussed in Q4) and extrapolation error, because the $Q$-function is reset according to the learned policy,
> > which significantly mitigates the over-estimation problem as shown in Section M.
> >
> >
> > [1] Kostrikov, Ilya, Ashvin Nair, and Sergey Levine. "Offline Reinforcement Learning with Implicit Q-Learning." International Conference on Learning Representations. 2021.
> >
> > [2] Bai, Chenjia, et al. "Pessimistic Bootstrapping for Uncertainty-Driven Offline Reinforcement Learning." International Conference on Learning Representations. 2021.
> >
> > [3] Ajay, Anurag, et al. "OPAL: Offline Primitive Discovery for Accelerating Offline Reinforcement Learning." International Conference on Learning Representations. 2020.
> >
> > [4] Lee, Jongmin, et al. "Optidice: Offline policy optimization via stationary distribution correction estimation." International Conference on Machine Learning. PMLR, 2021.
> >
> > [5] Scott Fujimoto, David Meger, Doina Precup. ``Off-Policy Deep Reinforcement Learning without Exploration.'' International Conference on Machine Learning. PMLR, 2019.
> >
> > [6] Tarasov, Denis, et al. "CORL: Research-oriented Deep Offline Reinforcement Learning Library." arXiv preprint arXiv:2210.07105 (2022).

---

### Official Review · Reviewer_mP3E · 2022-11-01

**Confidence:** 3
**Correctness:** 3
**Technical Novelty And Significance:** 4
**Empirical Novelty And Significance:** 2
**Recommendation:** 5

**Clarity, Quality, Novelty And Reproducibility:**

Clarity:
- Well-written text.
- Mathematical expressions are quite clear but provide some details that are superfluous, e.g.:
  * You could safely remove all occurrences of the environment discount d.
  * The detail of clipped double Q and Z is somewhat unnecessary and distracting. Could probably be stated and the exact update given in the appendix.
- Figure 4 has ablations that only make sense when looking at Appendix E, which isn't ideal.

Quality:
- Experiments in the paper are quite thorough.
- I like Figure 1 in principle, but it's unclear whether the top and bottom row are for a single sample state each, in which case it becomes anecdotal evidence. Could the authors find a way to produce a similar plot averaged over all states in/out of the offline dataset?
- From Figures 2 and 3, it's difficult to tell whether the method is helping much except for the datasets which include expert data.
- The tricks in Appendix A.2 seem quite significant and are hopefully used in the baselines as well.

Novelty: I haven't seen such an alignment step used to correct the Q-value extrapolation issue in offline pretraining. It's a neat idea.

Reproducibility: Given that the offline datasets and the authors' code is opensourced, I'm adequately satisfied regarding the reproducibility of this work.

**Strength And Weaknesses:**

Strengths:
- A nice and simple idea.
- Attacks an important and well motivated.
- The first third of the paper is quite clearly written.

Weaknesses:
- Clarity could be improved in parts (e.g. beta-clipping, discounts).
- Figure 1 could be made more rigorous.
- Experiments are thorough but the plots are hard to read.
- Benefit of the approach is not clear empirically.
  * Improvements are only clear in the expert offline data regime.
- The additional implementational details in A.2 and Eq. 8 seem like a confounder. Are these used for all baselines as well?

**Summary Of The Paper:**

The authors propose a new approach for improving the transition between offline and online training of policies via RL. In actor-critic algorithms, the source of errors in the online phase comes from extrapolation errors in the pretrained Q-function. In the online phase, when the pretrained policy gets into states that are not in the offline dataset, the Q-function can overestimate the values of poor actions and then the policy optimization in turn deteriorates the policy. The proposal in this paper is to fix this by throwing away the trained Q-function and effectively creating a new one that is "aligned" with the policy in the sense that the Q-values are fit to the log-policy. This is also somewhat theoretically justified by the common interpretation of policies being Boltzmann distributions with the Q-function as the energy.

**Summary Of The Review:**

Overall I think this paper introduces a neat and simple idea for offline-to-online RL. It's an area of great significance for real world RL applications. Unfortunately, the exposition is slightly lacking, particularly empirically. It seems the benefits of the method are really only significant when the dataset has a lot of expert trajectories. Given the alignment nature of the idea, I worry that this is a built-in weakness [1] and I'm not yet convinced by the experimental results.

I was an emergency reviewer so I will drop my confidence score to reflect that.

[1] The authors even state: "the Q-function is generally problematic for out-of-distribution actions while the policy learned offline is assumed trustworthy."

---

> ### Author Response · Authors · 2022-11-18
> **Subject: Response to Reviewer mP3E**
>
> We thank the reviewer for the constructive review. We provide a detailed response to your concerns below.
>
> **Q1:**  Figure 1
>
> **A:** We give similar plots in Section M for in-distribution samples and out-of-distribution samples. We also include CQL and AWAC for comparison rather than using SAC+ML alone as baseline.
>
> In Figure 14 and Figure 15, we measured how many $(s, a)$ pairs exhibit the behavior that $Q(s, a) > Q(s, \pi(s))$, which reflects to some extent how the $Q$-function "disagrees" with $\pi$. We refer to it as "over-estimation".
> Section M details how these figures are generated.
>
> ACA achieves the least "over-estimation" for both in-distribution and out-of distribution data. CQL comes as a runner-up, much stronger than SAC+ML and AWAC, which matches the empirical performance shown in Section 6.1. However, CQL still suffers a much larger fractions of actions that are over-estimated when facing out-of-distribution samples.
>
>
>
> **Q2:** Plots are hard to read
>
> **A:** We updated the caption of Figure 2 to refer to legend defined in Table 1, which makes it easier to connect the methods with the labels, and the caption of Figure 3 to make it is easier to interpret. Figure 4 has also been updated to be more self-contained.
>
>
>
> **Q3:** Is the method helping much except for the datasets which include expert data?
>
> **A:** Table below shows the results after excluding medium-expert and expert.  The final performance is only marginally lower than balanced replay but our performance increase is higher. Overall we would safely argue that our approach **matches** BR in performance,
> hence highlighting both BR and ours in Table 2.
> That said, we would like to reiterate that our method enjoys a significant advantage of attaining nearly the same performance without re-accessing offline dataset, a scenario where balanced replay is not even applicable.
>
>
> | Dataset     | Env         | Score($\delta$)  CQL$\to$BR | Score($\delta$) SAC$\to$ACA |
> |-------------|-------------|----------------------------------|-----------------------------------|
> |             | HalfCheetah | 84.36(59.06)                     | 72.60(55.31)                      |
> | Random      | Hopper      | 29.80(29.15)                     | 81.85(73.92)                      |
> |             | Walker2d    | 10.05(9.39)                      | 12.42(9.06)                       |
> |             | HalfCheetah | 82.95(34.52)                     | 66.58(20.25)                      |
> | Medium      | Hopper      | 98.14(24.67)                     | 96.54(42.24)                      |
> |             | Walker2d    | 76.36(-6.11)                     | 74.66(-6.50)                      |
> |             | HalfCheetah | 78.36(32.46)                     | 59.03(16.50)                      |
> | Med.-Replay | Hopper      | 97.25(1.28)                      | 85.54(36.72)                      |
> |             | Walker2d    | 100.06(21.68)                    | 85.17(22.98)                      |
> | Total       |             | 657.33(206.1)                | 634.39(270.48)                |
>
>
>
> Besides, another advantage over BR is that ours can be initialized from a broader choice of offline RL algorithms, as shown in Section 6.3. BR's performance relies critically on CQL offline training while ours can be initialized from different choices.
>
> In addition, we designed a new set of experiment to study the scenario where only the actor is updated, on out-of-distribution data, excluding other factors such as critic update and data collecting.  In Section K, we found that it is much harder to attain the offline policy performance for expert-level tasks, which suggests that near optimal policies are even more fragile. Therefore, consistent performance in expert-level should be regarded as an advantage instead of weakness.
> In this experiment, our re-parameterized $Q$-functions also did the best job to attain offline policies' performance, which verifies our motivation underlying the choice of Eqn (5). It also verified that ours can be initialized from different offline algorithms rather than relying critically on a particular one.
>
>
> **Q4:** Clarification about tricks
>
>  **A:**
> The tricks in Appendix A.2 only applies to the
>  $\log \pi_0$ term,
>  which is not present in other baseline methods.
>  The trick in Eqn (8) is designed by TD3+BC to balance actor-critic and behavior cloning.
>  Thus there is no need to include it in CQL or other baselines.

---

### Official Review · Reviewer_LK37 · 2022-11-03

**Confidence:** 4
**Correctness:** 3
**Technical Novelty And Significance:** 3
**Empirical Novelty And Significance:** 2
**Recommendation:** 6

**Clarity, Quality, Novelty And Reproducibility:**

* The paper is well-written barring a few interpretation difficulties illustrated in the earlier sections.

* There are two key original ideas in the work -- disregarding Q values and aligning them with policies learned offline, reconstructing Q values from the policy and state-only values during the online fine-tuning phase.

* The authors have also shared fair amount of details about their experiment settings and code, thus encouraging reproducibility of the experiments which were run in environments that are extensively studied in literature.

**Strength And Weaknesses:**

Strengths

* The approach to disregard learned Q values during the alignment phase and aligning them into the warmstart policy and state-only function is an interesting idea for transfer from offline to offline settings.
* The paper does some good ablations (eg Figure 1 and Figure 4) where they first justify the alignment procedure and later verify that BC regularization alone does not guarantee offline -> online stability, both of which are very valuable observations.
* The approach does not require any of the offline data during online phase, which can be suitable for applications that are covered by data privacy considerations such as healthcare.
* The results are strong and consistent over the environments taken into consideration while all other baselines fail to transfer well in some cases or the other.


Areas to improve/ Clarifications and Questions

* Should we also consider offline RL approaches such as the model-based ones which learn the dynamics structure before the fine-tuning phase for a fair comparison to how they respond to OOD actions? Running full-scale experiments might be a costly exercise, but I still feel additional experiments or some discussion around this aspect can advise which approaches to embrace for offline-online RL.
* Transferring Q values to policies might sometimes swing the pendulum to the other side from over-optimistic estimates to pessimistic estimates in the policy, which may prevent exploration in the online fine-tuning case. Prior works (eg [2]) have alluded to this phenomenon. It might be worth investigating whether such a phenomenon affects the model from learning more optimal policies in certain environments where it underperforms other baselines.
* (please correct me if i misunderstand) The alignment procedure works only when the Q function estimates the values of the behavior policy, but I am not sure whether this transfer is possible for methods that do perform multi-step DP on the value functions such as Implicit Q Learning [1]. It might be worth having a discussion about this specific aspect in the paper and see how this transfer might work for those value functions.
* The paper uses the last checkpoint during offline training as the initialization for online training. This might not always be the best offline trained model as it does not account for overfitting and thus, exacerbating the offline -> online OOD effect even more. Since this is a study in controlled environments, policy selection could still be done by evaluating in the environment and picking the best possible policy to warmstart. This can expose whether overfitting might be a strong reason for the models to suffer in terms of generalizing to the online environment.
* Minor : The paper could have benefitted from demonstrating transfer from offline to online in more complicated real-life environments plagued by distribution shift, such as robotics where sample collection is costly. This would have made the impact of results even stronger.
* Minor:  For baselines in transfer such as SAC -> SAC, while distribution shift is a problem, an interesting ablation would be to constrain the policies from diverging too much during the online training at each training step such as adding KL penalties for policy divergence, for instance. This would further explain whether we need this alignment step as compared to controlling policy divergence in the online phase from the bootstrapped policy.

Minor comments

Figure 2 might be useful to point users to table 1 to understand the legends better.

Page 2 - Our approach does not “reply” on should have rather been rely on

Exposition in figure 1 is not clear. Kindly amend the description to better demonstrate how the alignment is benefitting over the baseline.

Figure 4 can be made more self-contained as ablation 1 and ablation 2 are not really clear anyone trying to parse the figures.

References
​​1. Implicit Q Learning – https://arxiv.org/pdf/2110.06169.pdf

2. https://arxiv.org/abs/2002.12174



**Summary Of The Paper:**

The paper proposes a novel alignment step for actor-critic RL approaches from offline to online settings. The paper relies on the observations that the Q functions OOD are much more untrustworthy than the entropy regularized policies learned from the offline data. They propose to disregard Q-values learned offline to combat overestimation under distribution shift and rather propose a procedure to rebuild these Q functions from the policy and a state-only value estimate during the online fine-tuning phase. They demonstrate little to no drop in performance when offline to online transfer happens on a range of controlled simulated tasks.

**Summary Of The Review:**

While the results are not significantly better than a strong competitive baseline CQL + BR, the ideas of aligning Q values from offline to online phase and reconstructing Q values for online finetuning are fairly interesting. The experiments have been run on limited  environments and not in large domains where the performance shift arising out of the distribution shift can be very evident. The paper has done a fair job of running ablations and demonstrating which components of the alignment step are important for the transfer. While there are certain clarifications highlighted earlier, in the interest of the idea, I am recommending to marginally accept the paper and hope that the authors do address the comments in subsequent iterations.

---

> ### Author Response · Authors · 2022-11-18
> **Subject: Response to Reviewer LK37**
>
> We thank the reviewer for the constructive review. We provide a detailed response to your concerns below.
>
> **Q1:** Model-based RL
>
> **A:**
> Although model-based approaches seem appealing as the MDP does not change during the transfer, it has been noted that they also suffer from the distribution shift in state marginals and actions (Mao et al., 2022; Kidambi et al., 2020; Yu et al., 2020; Janner et al., 2019). They may exploit the model to pursue out-of-distribution states and actions where the model mis-believes to yield a high return.
>
> As a result, delicate efforts are required to place the comparison on an equal footing.  We therefore refrained from comparing with model-based methods, in the same way as the papers of AWAC, Balanced Replay, ODT, IQL.
>
>
>
> **Q2:** Pessimistic estimates
>
> **A:**
> Yes, we do observe that in some tasks where the initial performance was not hurt much (e.g., halfcheetah-medium), our method is slightly slower than other baselines in improving the performance.
> Since our focus is on *safer transfer*, we consider this as an inevitable trade-off.
>
> However, this should not prevent our agent from eventually learning a better policy, because we do not force the $Q$-function to stay on the closed-form manifold in the entire process of online training; see Section 4.3.
> Indeed, we only **initialize** $R(s, a)$ with $Z(s)$ rather than keeping it on this manifold. So the impact of our initialization diminishes as the online learning progresses.
>
> For the purpose of demonstration, we ran our method for 500k online steps on halfcheetah-medium where the effect of over-pessimism appears most evident. The detailed results can be found in Section Q.
> In particular, we experimented with a less conservative setup with a lower value of $k$ in front of the log-likelihood term in Eq (28).
> In Figure 20, the original hyper-parameter values (blue line) lead to a near-optimal policy w.r.t. D4RL expert score. In contrast, the more optimistic one (green line), with no surprise, converges to the D4RL expert level performance.
> This demonstrates that the hyper-parameters, which we did not tune exhaustively, leave good room of improvement by being more optimistic, which is consistent with the theory and observations made by the existing literature.
>
>
>
> **Q3:** Multi-step DP
>
> **A:** Our alignment step readily applies to multi-step DP.
> In fact, we have done the experiments for CQL.
> Note that  both CQL and TD3+BC are considered as multi-step DP according to IQL (Kostrikov et al., 2022b).
>
>
>
>
> **Q4:** Initialization from best checkpoints
>
> **A:** We believe evaluating offline RL performance is still a difficult research problem, and using the last checkpoints should be practical. However, as it was implemented in a controllable environment, we ran the experiments as the reviewer suggested, with results given in Section O. We do not observe significant difference between initialization from best or last checkpoints.
>
> To further investigate whether overfitting is an issue, we took CQL trained on walker2d-medium as an example. As CQL converged earlier enough and showed significant drop in online stage, this would be a good case for studying the effect of overfitting. In Section P, we ran transfer with checkpoints at 100k, 200k, 300k, 400k and compared with the original result from 500k. There is no evidence that overfitting is an issue here.
>
>
> **Q5:** Real-life/robotics tasks
>
> **A:** Unfortunately we were unable to implement relevant experiments as we do not have access to any physical robots.
>
>
> **Q6:** Adding KL constraints
>
> **A:** We found AWAC should somehow reflect this concern as AWAC is based on KL constraints. Results in Table 2 have shown our advantage over AWAC. In addition, we implemented an additional experiment which examines how out-of-distribution samples could affect AWAC actor updates. This experiment ran AWAC actor updates only on out-of-distribution data, excluding other factors such as critic update and data collecting. It can be observed that AWAC's performance collapses rapidly. The details can be found in Section L. In contrast, Section K shows that our alignment could better attain the offline performance while running actor updates on out-of-distribution data.
>
>
> **Q7:** Figures and typo
>
> **A:** We have updated captions of Figure 2 and Figure 4, and fixed typo accordingly.

---

### Decision · Program_Chairs · 2023-01-20

**Decision:**

Reject

**Justification For Why Not Higher Score:**

Noted in the decision part of the meta-review. The paper requires a major revision before acceptance.

**Justification For Why Not Lower Score:**

N/A

**Metareview: Summary, Strengths And Weaknesses:**

### Summary
Offline RL can significantly reduce the amount of online interactions by leveraging abundant offline data. However, the benefits of pretraining with Offline RL diminishes with the state-action distribution shift which can cause bootstrapping errors. This paper proposes an approach that addresses the problem of distribution shift that happens when transitioning into online RL from offline RL. The paper shows empirical results on three environments from the datasets D4RL-v2, including HalfCheetah, Hopper, and Walker2d.

In the strengths and the weaknesses, I will be  summarising the points reviewers have made during the rebuttal period:

### Strengths

- Studying an important problem with an interesting approach.
- Strong and convincing experimental results.

### Weaknesses
- Clarity could be improved in parts (e.g. beta-clipping, discounts) but also there are some terms that are not clearly explained such as 'actor-critic misalignment'.
- Experiments are thorough but the plots are hard to read. More ablations are needed on possibly smaller/synthetic tasks where different parts of the algorithm can be controlled more carefully.
- It is hard to follow the writing and understand the motivation for some choices.

### Decision

This paper attacks a vital problem to get RL into real-world problems. The proposed approach is interesting, and the scores of this paper were borderline. The authors have done an excellent job during the rebuttal presenting new results to address the concerns raised by the reviewers. However, the paper still has many fixes to be done before acceptance. The writing of the paper requires some improvements, for example, there are unclear statements like "marked state-action distribution" and "actor-critic misalignment." I recommend the authors be more rigorous in explaining these concepts carefully. I think this paper would benefit from further ablations, possibly in simpler and more toyish environments. Most experiments focus on the datasets coming from experts or near-experts. Ablations on the quality of the data and the effect on the performance of the online finetuning would be useful. All those changes require significant revision of the original paper. I recommend the authors make the required revision and strongly encourage a future submission to another ML venue.




**Summary Of Ac-Reviewer Meeting:**

I have had a chat on this paper with the Reviewer LK37 and they were the reviewer that gave the highest score for this paper. The reviewer mentioned about the strengths and weaknesses of the paper. The reviewer mentioned that they like the idea but the paper is written poorly and lacking ablations. They were on the positive side, because they thought the idea was good and recommended the reviewers to consider submitting to a future venue with the changes recommended by the reviewers.